# Defective mitochondrial COX1 translation due to loss of COX14 function triggers ROS-induced inflammation in mouse liver

Abhishek Aich [1,2], Angela Boshnakovska [1], Steffen Witte[1], Tanja Gall[1], Kerstin Unthan-Fechner[3], Roya Yousefi[1], Arpita Chowdhury [1], Drishan Dahal[1], Aditi Methi[4,5], Svenja Kaufmann[6], Ivan Silbern [6,7], Jan Prochazka [8], Zuzana Nichtova [8], Marcela Palkova[8], Miles Raishbrook[8], Gizela Koubkova [8], Radislav Sedlacek [8], Simon E. Tröder [9], Branko Zevnik[9], Dietmar Riedel [10], Susann Michanski[2,11], Wiebke Möbius [11], Philipp Ströbel[12], Christian Lüchtenborg[13], Patrick Giavalisco [14], Henning Urlaub [2,6,7], Andre Fischer[2,4,5,15], Britta Brügger [13], Stefan Jakobs [2,16,17,18] & Peter Rehling [1,2,18,19] ✉

Mitochondrial oxidative phosphorylation (OXPHOS) fuels cellular ATP demands. OXPHOS defects lead to severe human disorders with unexplained tissue specific pathologies. Mitochondrial gene expression is essential for OXPHOS biogenesis since core subunits of the complexes are mitochondrial-encoded. COX14 is required for translation of COX1, the central mitochondrial-encoded subunit of complex IV. Here we describe a COX14 mutant mouse corresponding to a patient with complex IV deficiency. COX14[M19I] mice display broad tissue-specific pathologies. A hallmark phenotype is severe liver inflammation linked to release of mitochondrial RNA into the cytosol sensed by RIG-1 pathway. We find that mitochondrial RNA release is triggered by increased reactive oxygen species production in the deficiency of complex IV. Additionally, we describe a COA3[Y72C] mouse, affected in an assembly factor that cooperates with COX14 in early COX1 biogenesis, which displays a similar yet milder inflammatory phenotype. Our study provides insight into a link between defective mitochondrial gene expression and tissue-specific inflammation.

Mitochondria enable eukaryotic cells to efficiently utilize the chemical energy released from oxidation of nutrients. The mitochondrial oxidative phosphorylation system produces the bulk of cellular ATP. The four respiratory chain complexes are membrane bound enzymes that catalyze redox reactions, directing electrons from NADH and $FADH_2$ to molecular oxygen to produce water. This process establishes a proton gradient that drives synthesis of ATP by the $F_1F_o$ ATP-synthase. Complexes I, III, IV, and V consist of subunits of dual genetic origin, encoded by both mitochondrial and nuclear genomes. Exceptional spatial and temporal coordination between different pathways of protein expression, transport, and assembly are required to enable proper biogenesis of these complexes[1–3].

In the case of cytochrome *c* oxidase (complex IV), the terminal enzyme of the respiratory chain, the mitochondrial-encoded COX1 subunit represents the headstone of the biogenesis process. COX1 is cotranslationally inserted into the inner mitochondrial membrane with the aid of the OXA1L insertase[4–6]. Already during synthesis, COX1 engages with proteins of an assembly intermediate termed

MITRAC, which enables engagement of COX1 with nuclear-encoded subunits[7]. The early MITRAC subunits C12ORF62 (COX14) and MITRAC12 (COA3) associate with the ribosomes-COX1 nascent chain complex to assist in COX1 membrane insertion and stabilization of the nascent polypeptide. During the assembly process, COX1 passes through a series of assembly stages during which additional mitochondrial-encoded and nuclear-encoded proteins join to form the complex IV. These intermediates are stabilized by stage-specific chaperone-like assembly factors[8]. Interestingly, COX1 translation depends on the influx of nuclear-encoded complex subunits to adapt mitochondrial gene expression to cellular demands[9]. The cytochrome *c* oxidase is equipped with multiple nuclear-encoded tissue-specific isoforms for numerous subunits[10]. Since their function is still poorly understood, they are of broad interest in the field. It has been hypothesized that different COX isoforms regulate the activity and structure of the enzyme to meet different energy demands of distinct tissues[11]. Dysfunction of either structural or non-structural proteins associated with complex IV biogenesis have been linked to severe human disorders with a surprising variability of effects on different tissues[12]. This makes the comprehension of the pathophysiology challenging.

In human mitochondria, C12ORF62 (COX14) and MITRAC12 (COA3) have been shown to stimulate the translation of COX1 in a ribosome nascent chain complex. In the absence of either protein, translation of COX1 is blocked leading to a loss of complex IV. Patients affected in both genes have been identified. The patient carrying a M[19]I substitution in C12ORF62 exhibited severe metabolic acidosis and death in the neonatal period[13]. All major organ systems, prominently brain hypertrophy, hepatomegaly, hypertrophic cardiomyopathy, renal hypoplasia, and adrenal-gland hyperplasia were observed. Identical disease presentation was later observed in the two subsequent siblings in the same family. Biochemical analysis of patient fibroblasts showed reduced complex IV activity and COX1 translation. On the other hand, the patient carrying a Y[72]C exchange in MITRAC12 showed a mild clinical course despite strong reductions in complex IV amounts[14]. The most severe symptoms were exercise intolerance and peripheral neuropathy. It is not clear how defects in the same assembly pathway leading to similar losses of complex IV display different pathological manifestations.

Here we established two mice models, COX14[M19I] and COA3[Y72C], recapitulating the amino acid exchange observed in the patients. The COX14[M19I] mouse displayed multi-systemic pathology that worsened with age. Different magnitudes of gene expression defects across tissues could be attributed to differential stability of the mutant COX14. Despite the common defect in COX1 translation, the resulting complex IV deficiency and tissue pathology varied greatly. Interestingly, in liver, which displayed the highest reduction in complex IV, a strong inflammation was apparent. Furthermore, we observed an increased contact between mitochondria and lipid bodies in the mutant liver. Our data revealed that mitochondrial RNA release into the cytoplasm of the hepatocytes activates a mitochondrial RNA triggered Type I IFN response and local inflammation. Mitochondrial RNA release is triggered by increased mitochondrial ROS production in mutant mitochondria due to loss of complex IV. Accordingly, we reveal a pathology in which increased ROS production correlating with the loss of complex IV triggers mitochondrial RNA release and concomitant induction of inflammation pathways that contributes to hepatic failure.

## Results

### COX14[M19I] leads to protein instability affecting complex IV assembly

COX14 (C12ORF62 in human) mediates early steps of cytochrome *c* oxidase (complex IV) assembly by coordinating translation of mitochondrial COX1 with early biogenesis steps. In order to analyze the molecular pathology of loss of COX14 we generated a mouse model by CRISPR/Cas-mediated patient mutation knock-in. We introduced a missense point mutation G88A into the *Cox14* allele. This resulted in a single amino exchange at position 19 from methionine to isoleucine (Fig. 1a, Supplementary Fig. 1a). We were able to generate homozygous mice, COX14[M19I], which were used in the entire study. The altered nucleotide sequence in the *Cox14* mRNAs did not compromise its stability across the different tissues tested (Fig. 1b). However, the mutant protein was detected to varying degrees in different organs of the mice (Fig. 1c, Supplementary Fig. 1b). In agreement, translation of COX1, assessed by *in organello* [$^{35}$S] methionine labelling of mitochondrial translation products, was significantly affected in mutant mitochondria (Fig. 1d). Subsequent pulse chase experiments showed a slight effect on the stability of the newly synthesized COX1 compared to the wild type situation (Fig. 1e). Thus, COX14[M19I] apparently affected both translation and stability of COX1.

In steady state protein level analyses, COX14[M19I] mitochondria displayed a decrease in both nuclear- and mitochondrial-encoded cytochrome *c* oxidase subunits (Fig. 1f). Moreover, complex IV activity and amount were significantly reduced in mutant mitochondria isolated from different tissues (Fig. 1g, Supplementary Fig. 1c). Among the tested tissues, liver appeared to be the most affected tissue. Furthermore, COX14[M19I] liver mitochondria displayed a significant reduction in respiration (Fig. 1h). Hence, COX14[M19I] mice allowed a detailed dissection of complex IV biogenesis defects in tissues of mice. In addition, we identified tissue-specific effects of the COX14[M19I] mutation on COX1 translation and assembly.

### Phenotyping COX14[M19I] pathology

To assess pathologies of COX14[M19I], the mice were subjected to a standard workflow following the international mouse phenotyping consortium (IMPC) pipeline. The animals exhibited a slight decrease in body weight after birth. Upon aging, the effect was only retained in COX14[M19I] males which displayed a significant reduction when measured at 30 weeks of age (Fig. 2a). Moreover, 26% of the COX14[M19I] animals experienced pre-weaning mortality as compared to that in the wild type. Once the animals were weaned, the life expectancies were quite similar in males whereas the females lived a shorter lifespan as compared to that of the wild type (Fig. 2b, Supplementary Fig. 2a). Interestingly, various blood parameters were altered in the COX14[M19I] mice (Fig. 2c and Supplementary Data 1). Notably, we found elevated levels of alanine aminotransferase (ALT) and aspartate aminotransferase (AST), indicative of liver damage in mutant mice. Additionally, serum cholesterol and creatinine levels were high, suggesting dysfunctions of liver and kidney.

In recent years, mitochondrial dysfunction has been shown to affect development and function of immune cells[15]. We identified differences in several immune cell populations in the spleen of COX14[M19I] animals (Fig. 2d). Resting immune cell populations such as the resting CD8 + T cells, resting CD4- NKT cells and resting NK cells were found to be increased. In contrast, effector NK cells and effector CD4- NKT cells were decreased. This suggested either a metabolic deficit to fuel the activation or exit of certain populations from spleen. However, no prominent structural changes were observed in spleen sections.

Histological analysis of heart, skeletal muscle, liver, and eye indicated that these tissues were affected by the COX14 mutation (Fig. 2e). COX14[M19I] heart sections identified groups of vacuolated and fragmented cardiomyocytes. In some sections, hypertrophic cardiomyocytes with enlarged nuclei were observed in the left ventricle. Liver sections displayed spots of focal necrosis demarcated by mononuclear cell infiltrate. The skeletal muscle sections showed few areas of fibers with more basophilic central nuclei indicating localized myopathy and regeneration. While COX14[M19I] eye sections resembled the wild type at 16 weeks; at 30 weeks of age, thinning of the outer nuclear layer (ONL) was apparent in all sections, indicative of a lower number of

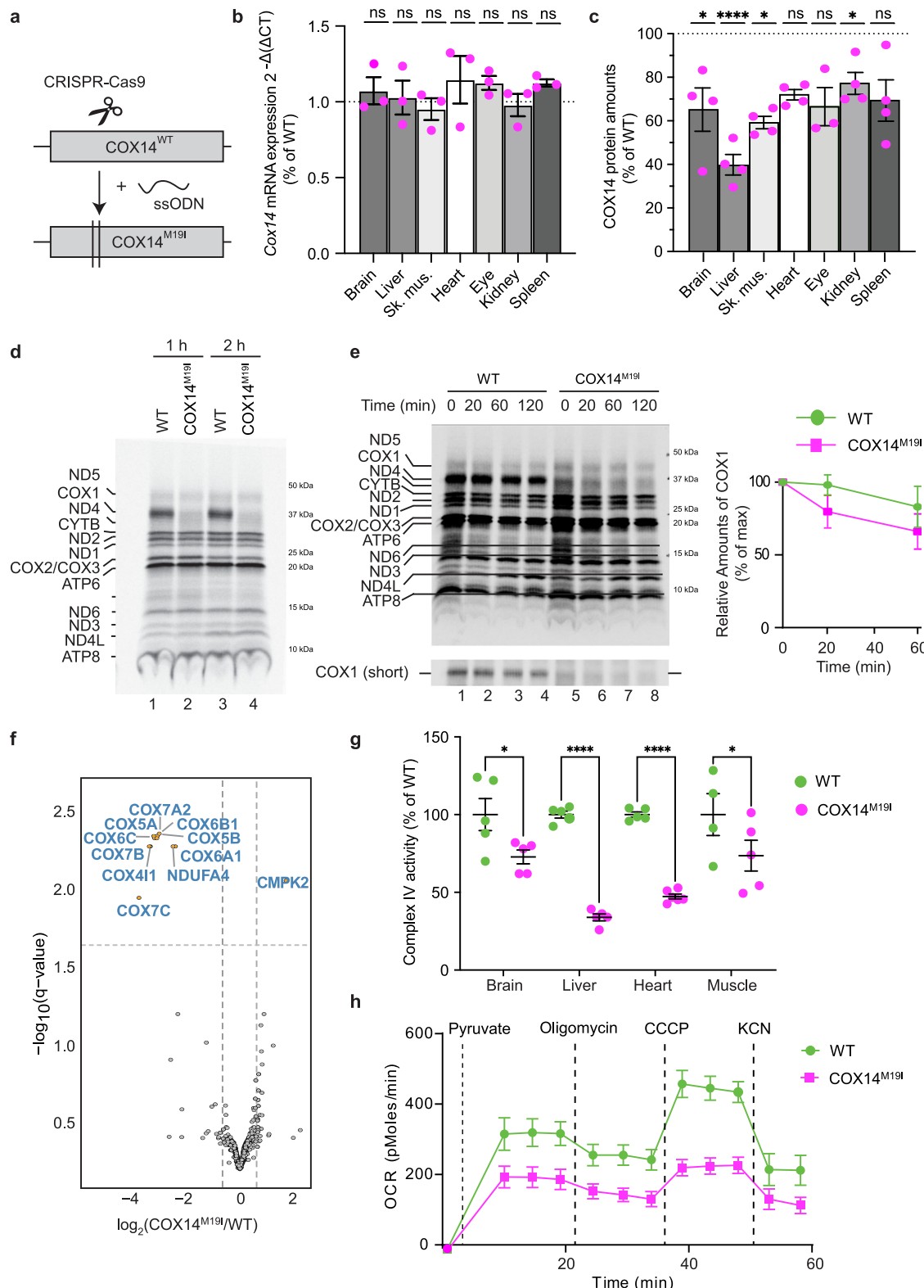

photoreceptors cells. In 25% of the animals, the outer plexiform layer (OPL), the interface of bipolar cells and photoreceptors, was also found to be reduced in thickness. In addition, optical coherence tomography revealed morphological changes in the vasculature of the eye (Fig. 2f, Supplementary Fig. 2b). The retinal vessels were tortuous, branched, and bent in the superficial vascular plexus. Furthermore, a decrease in retina thickness was observed for the COX14[M19I] animals with age, starting at 5 weeks till the last measurements at 30 weeks

(Fig. 2g). Analysis of the phototransduction by electroretinography showed a defect specifically in rod cells of the retina (Supplementary Fig. 2c). The COX14[M19I] retinas also exhibited increased positive TUNEL signal in the outer nuclear layer (ONL) and the inner nuclear layer (INL) in animals already at 15 weeks of age (Supplementary Fig. 2d) indicating a gradual degeneration of the retina with age.

Since the heart sections showed indications of hypertrophy, the animals were subjected to echocardiography. Similar to the results

**Fig. 1 | COX14$^{M19I}$ mice display reduced complex IV amounts. a** Cartoon depicting the generation of COX14$^{M19I}$ mouse using CRISPR/CAS9 mediated double stranded cut in *Cox14* allele and subsequent single stranded Oligonucleotide DNA (ssODN) mediated repair. **b** Expression of *Cox14* mRNA in indicated tissues of 35-week-old COX14$^{M19I}$ mice compared to wild-type (WT). Means ± SEM, $n = 3$, unpaired *t* test, ns = non-significant. **c** Western blot analysis and quantification of relative amounts of COX14 protein in indicated tissues from 22-week-old COX14$^{M19I}$ mice (Supplementary Fig. 1b). Presented relative COX14 amount as percent of WT after normalization to RIESKE. Means ± SEM, $n = 4$, unpaired *t* test, Brain $p = 0.0175$; Liver $p < 0.0001$; Muscle $p = 0.0242$; Kidney $p = 0.0393$. **d** [$^{35}$S] methionine labeling of mitochondrial translation products in isolated brain mitochondria from WT and COX14$^{M19I}$ mice. Samples were analyzed by SDS-PAGE and digital autoradiography The figure is representative of $n = 3$. **e** Mitochondrial translation products pulse labeled with [$^{35}$S] methionine for 1 h in WT and COX14$^{M19I}$ mice brain mitochondria and chased with non-radioactive methionine for indicated times. Samples were analyzed by SDS-PAGE and digital autoradiography. Quantification of COX1 presented. Means ± SEM, $n = 3$. **f** Volcano plot of mass spectrometric analysis of mitochondrial proteomes from WT and COX14$^{M19I}$ mice liver mitochondria, $n = 4$. **g** Measurement of cytochrome *c* oxidase activity from indicated tissues of 12-week-old WT and COX14$^{M19I}$ mice plotted as percentage of the average of WT. Means ± SEM, $n = 5$, One-way ANOVA, Brain $p = 0.0312$; Liver $p < 0.0001$; Heart $p < 0.0001$; Muscle $p = 0.0405$. **h** Real-time respirometry analysis of 35-week-old WT and COX14$^{M19I}$ mice liver mitochondria; oxygen consumption rates, OCR. Means ± SEM, $n = 3$. Source data are provided as a Source Data file.

obtained for the eye, no statistically significant changes were observed at 16 weeks of age. However, upon aging these animals to 30 weeks, an increase in the ventricular wall thickness and fractional area shortening were observed (Fig. 2h, Supplementary Fig. 2e). A concomitant decrease in the ventricular diameter was detected. Yet, the ejection was slightly increased in case of female mice but not significantly altered in male mice. All of these indicated an altered heart function with age. In conclusion, similar to the patients with Leighs syndrome-like disorders, arising from a similar mutation, COX14$^{M19I}$ newborn mice exhibited neonatal death (about 26%) and multisystemic pathologies that worsened with age.

### Transcriptome analysis reveals dysregulation of signaling pathways

Data on the cytochrome *c* oxidase, serum parameters, and histopathology showed that the liver was one of the most severely affected tissues in COX14$^{M19I}$ mice. To gain further insights into the key pathways affected, we addressed changes in the gene expression pattern by RNA sequencing (RNA-seq) of wild type and COX14$^{M19I}$ mice liver samples. An over-representation (pathway enrichment) analysis using terms from the KEGG pathway database indicated that pathways of respiratory electron transport chain, ATP synthesis, thermogenesis, plasma lipoprotein remodeling and chylomicron remodeling were downregulated in the mutant samples compared to the wild type (Fig. 3a). Conversely, pathways related to cholesterol biosynthesis, metabolism of steroids, antiviral IFN stimulated genes, class I MHC peptide loading and antigen presentation were enriched in COX14$^{M19I}$ livers (Fig. 3b). Differential gene expression analysis revealed an increase in several SERBP-dependent cholesterol biosynthesis genes including the 3-Hydroxy-3-methylglutaryl-CoA reductase (*Hmgcr*), Squalene epoxidase (*Sqle*), and Insulin induced gene 1 (*Insig1*) (Fig. 3c, d). Additionally, genes involved in the antiviral interferon alpha/beta pathway such as Interferon regulatory factor 7 (*Irf7*), Interferon alpha inducible protein 27 (*Ifi27*), Interferon induced protein with tetratricopeptide repeats 3 (*Ifit3*) and many more were found to be significantly upregulated in the context of COX14$^{M19I}$. Metabolic genes such glucose-6-phosphatase (*G6pc*); mitochondrial glutamate carrier, solute carrier family 25 member 22 (*Slc25a22*); and zinc transporter, solute carrier family 39 member 5 (*Slc39a5*) were found to be downregulated in COX14$^{M19I}$ liver samples. To gain insights into a causal map of molecular interactions, gene regulatory networks were built around the two highly upregulated transcripts *Irf7* (Fig. 3e) and *Hmgcr* (Supplementary Fig. 3). In case of *Irf7*, mostly second-degree contacts are perturbed which appear to be linked through the node of TRAF6, the mediator of RIG-I- and MDA5-mediated antiviral responses. However, the network of *Hmgcr* pointed to a dense network of first-degree contacts to be either upregulated or downregulated. Interestingly, many mitochondrial ribosomal proteins in this network map were downregulated. In summary, the investigation of alterations in gene expression revealed activation of inflammatory response pathways and indicated alterations in liver tissue metabolism.

### Aberrant lipid droplet association with mitochondria in COX14$^{M19I}$ livers

The transcriptome analysis of liver indicated upregulated cholesterol biosynthesis. This observation led us to investigate cellular lipid profiles by mass spectrometry-based lipidomics. Yet, these analyses identified no significant alterations in any of the identified lipid classes in COX14$^{M19I}$ liver (Fig. 4a and Supplementary Data 2). Furthermore, in order to substantiate these findings, primary COX14$^{M19I}$ hepatocytes were cultured and subjected to BODIPY staining to detect differences in intracellular lipid organization. The number of lipid droplets in COX14$^{M19I}$ hepatocytes was more in number than in the wild type (Fig. 4b, Supplementary Fig. 4a). Statistical analysis of lipid droplet size and number revealed a difference in lipid droplet number but not in the total lipid volume per cell (Fig. 4c). In addition, primary hepatocytes were subjected to transmission electron microscopy (TEM) analysis to assess lipid droplet distribution (Supplementary Fig. 4b). Interestingly, drastic alterations in mitochondrial morphology were observed. Mitochondrial cristae had lost their stacked organization in the COX14$^{M19I}$ hepatocytes and the mitochondrial outer membrane appeared to be noncontiguous (Supplementary Fig. 4b). To address this in tissue, liver sections were subjected to TEM analysis (Supplementary Fig. 4c). Although the mitochondria exhibited wild type-like membrane organization, in COX14$^{M19I}$ liver section we observed mitochondria that wrapped around a lipid droplet. Subsequent focused ion beam scanning electron microscopy (FIB-SEM) of liver samples demonstrated that, this phenomenon was frequently observed in every liver cell imaged from COX14$^{M19I}$. The mitochondria were in close contact with lipid droplets to varying degrees (Fig. 4d, Supplementary Fig. 4d). The FIB-SEM data set was subjected to visual inspection of a large portion of the cell volume of hepatocytes and reconstructed in 3D. We observed that the mitochondria wrapping the lipid droplets were not spatially isolated entities compartmentalized to a part of the cell but rather were present across the entire cell (Fig. 4e). The extent of mitochondria-lipid droplet contacts differed. In extreme cases the mitochondria engulfed the lipid droplet, while in some cases they just had a single point of contact (Supplementary Fig. 4e).

To discern the functional implication of this contact, free mitochondria termed cytoplasmic mitochondria and lipid droplet associated peridroplet mitochondria were isolated from mice livers (Supplementary Fig. 5a, b). We subjected both populations to specific substrate-based real time respirometry analysis. However, we did not observe any preferential substrate utilization (Supplementary Fig. 5c). We concluded that despite of the apparent lipid droplet reorganization that was observed in the COX14$^{M19I}$ mice the liver did not display a significant alteration in lipid metabolism.

### mtRNA release to the cytosol from mitochondria

The transcriptome analysis revealed that the antiviral interferon alpha/beta pathway was highly upregulated in COX14$^{M19I}$ liver. Typically, these type I IFN responses or Interferon Stimulated Genes (ISGs) are

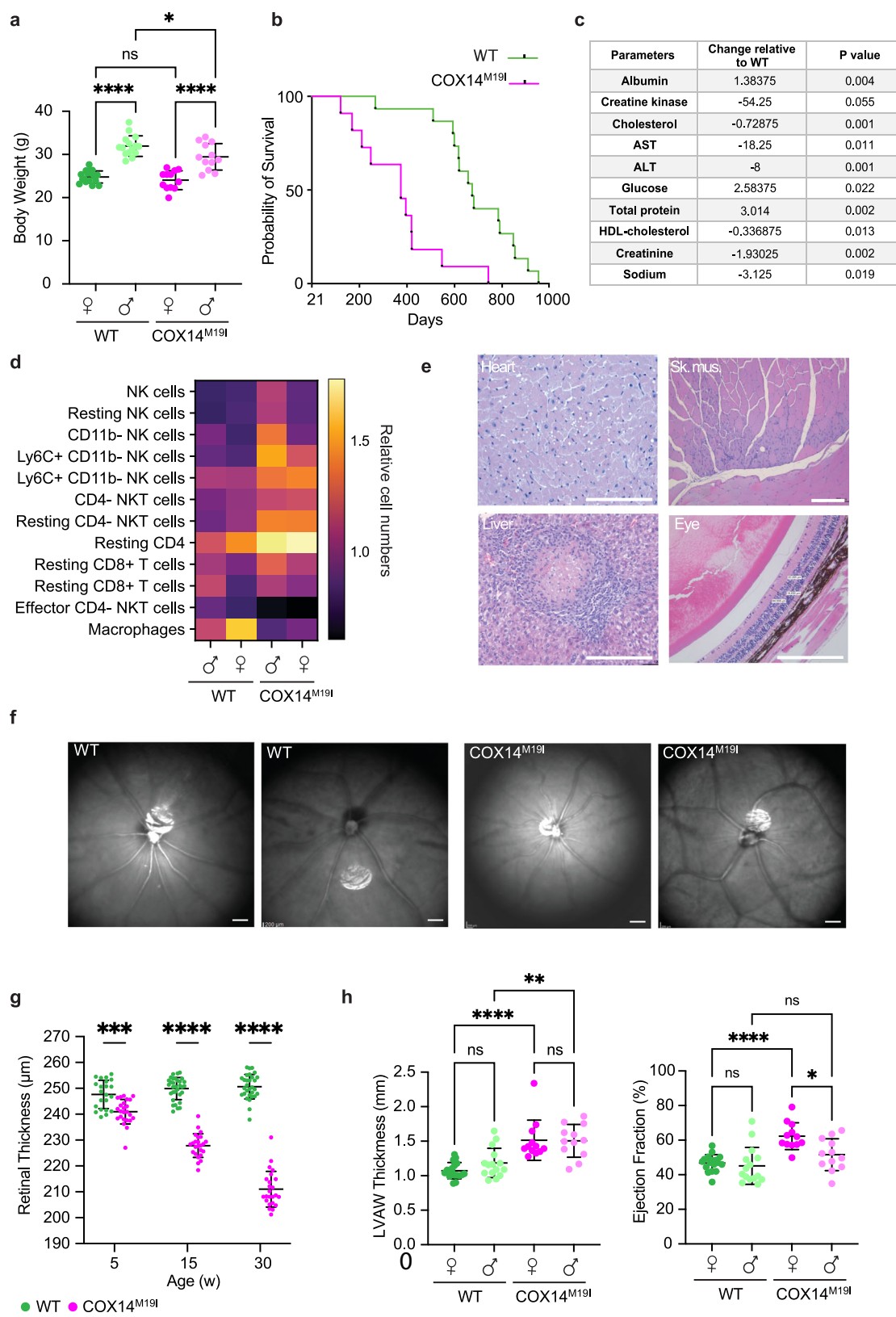

**c**

| Parameters | Change relative to WT | P value |
|---|---|---|
| Albumin | 1.38375 | 0.004 |
| Creatine kinase | -54.25 | 0.055 |
| Cholesterol | -0.72875 | 0.001 |
| AST | -18.25 | 0.011 |
| ALT | -8 | 0.001 |
| Glucose | 2.58375 | 0.022 |
| Total protein | 3.014 | 0.002 |
| HDL-cholesterol | -0.336875 | 0.013 |
| Creatinine | -1.93025 | 0.002 |
| Sodium | -3.125 | 0.019 |

expressed upon a wide variety of stimulants such as viral infections, cellular stress, ROS, or non-self-nucleic acid sensing[16–18]. COX14[M19I] liver showed a significant expression of ISGs both at RNA and protein levels (Fig. 5a, b). We observed that the mRNA expression of ISGs were also upregulated in other tissues such as the heart, skeletal muscle and brain but to a significantly lesser extent than in liver. In fact, the only other tissue to show a slight upregulation of ISGs at the protein level

was kidney (Supplementary Fig. 6a). Thus, liver was the only organ with a high type I IFN response.

To address if the observed inflammatory phenotype was specific to the COX14[M19I] mutation rather than a loss of complex IV activity, a second mouse model was generated using a similar strategy to target the functionally closely related gene COA3. Both COA3 (MITRAC12 in human) and COX14 act early in the COX1 assembly process and their

**Fig. 2 | Pathophysiology of COX14[M19I] mice. a** Body weights of wild-type (WT) and COX14[M19I] mice at 30 weeks of age. Means ± SD, $n = 12$, One-way ANOVA, ns non-significant, WT females vs. WT males $p < 0.0001$; COX14[M19I] females vs. COX14[M19I] males $p < 0.0001$; WT males vs. COX14[M19I] males $p = 0.0305$. **b** Survival curve of WT and COX14[M19I] female mice. **c** Serum biochemical parameters for 16-week-old COX14[M19I] mice compared to WT mice, $n = 8$, Unpaired $t$ test. **d** Heatmap depicting steady state immune cell populations from WT and COX14[M19I] mice spleen showing differences in ones with $p$-value < 0.05; One-way ANOVA; $n = 6$. **e** Representative H&E-stained sections from heart, skeletal muscle, liver and eye of COX14[M19I] mice. Scale bar 200 μm. The figure is representative of $n = 8$. **f** Representative fundus images from WT and COX14[M19I] eyes. Scale bar 400 μm. The figure is representative of $n = 8$. **g** Retina thickness WT and COX14[M19I] eyes at indicated age determined by Optical Coherence Tomography. Means ± SD, $n = 24$, One-way ANOVA, 5 weeks $p = 0.0005$; 15 weeks $p < 0.0001$; 30 weeks $p < 0.0001$. **h** Echocardiography (left panel, left ventricle anterior wall thickness; right panel, ejection fraction) of WT and COX14[M19I] mice. Means ± SD, $n = 12$, One-way ANOVA, ns=non-significant, left panel: WT females vs. COX14[M19I] females $p < 0.0001$; WT males vs. COX14[M19I] males $p = 0.0018$; right panel: WT females vs. COX14[M19I] females $p < 0.0001$; COX14[M19I] females vs. COX14[M19I] males $p = 0.0134$. Source data are provided as a Source Data file.

loss of function results in decreased COX1 translation[7]. A missense point mutation A583G was introduced into the *Coa3* allele. This resulted in the exchange of a single amino acid at position 72 from tyrosine to cysteine, similar to the amino acid exchange reported for a mitochondrial disease patient[14] (Fig. 5c, Supplementary Fig. 6b). COA3[Y72C] homozygous males and females did not display differences in life expectancies compared to the wild type (Supplementary Fig. 6d). COA3[Y72C] sera displayed significant differences in HDL and total cholesterol levels (Supplementary Fig. 6c). Since ALT and AST levels in the serum were similar to the wild type and no morphological differences were observed by histochemistry, we concluded that the livers from COA3[Y72C] mice did not display any apparent pathology. However, liver mitochondria showed a clear reduction in complex IV activity (Fig. 5d). Yet, the reduction in complex IV was not to the same extent as observed in COX14[M19I] mice. Moreover, the expression of ISGs was upregulated in the COA3[Y72C] mice but again to a lesser degree than in COX14[M19I] (Fig. 5e). We concluded that the observed inflammatory pathology was specific to COX14[M19I] mice and correlated with the defect on complex IV.

Apart from viral infections, type I IFN inflammation is also induced by cytosolically sensed mitochondrial nucleic acids (Supplementary Fig. 7a)[18,19]. Therefore, we analyzed if mitochondrial nucleic acids were present in the cytoplasm of COX14[M19I] liver. Liver tissue homogenates were cleared of mitochondrial and nuclear fractions and subjected to RNA and DNA isolation. We detected both mitochondrial RNA (mtRNA) and mitochondrial DNA (mtDNA) by qPCR in significant amounts in cytoplasmic fractions of liver cells (Fig. 5f and Supplementary Fig. 6e). The presence of mtDNA in the cytoplasm in other organs such as heart, brain, and skeletal muscle was not increased above wild type levels. Accordingly, among the organs tested, liver was the only one to show a significant release of mitochondrial nucleic acids into the cytoplasm.

To exclude that the observed detection of mitochondrial DNA was caused by nuclear-embedded mitochondrial DNA sequences (NUMTs) that were amplified by qPCR while quantifying mtDNA in the cytoplasm, an additional method was applied. Long-range multi-gene spanning probes covering three different regions of mouse mtDNA were designed to uniquely amplify the mtDNA (see Methods). These analyses further confirmed the presence of mtDNA in the cytosol of COX14[M19I] mice liver cells.

Based on the presence of mtDNA and mtRNA in the cytoplasm, we assessed which of the cytoplasmic sensors were involved in the detection of these nucleic acids. We found that RNA sensors such RIG-1 and MDA5 were significantly more abundant in liver cells than DNA sensors such as STING (Fig. 5g). Additionally, we also observe increase of the ZBP1 protein in the COX14[M19I] mice samples. Thus, we concluded that in COX14[M19I] mutant mice mitochondria released nucleic acids into the cytosol and thereby trigger a mtRNA mediated type I IFN inflammation in liver.

## ROS imbalance leads to mitochondrial nucleic acid release

Cellular pyrimidine deficiencies have been shown to trigger mtDNA-dependent immune responses in mouse retina and cells lacking the mitochondrial protease YME1L[20]. Hence a quantitative metabolomic screen with a focus on nucleotides, glycolysis, and TCA cycle intermediates was performed. Contrary to the case of YME1L deficiency, where a broad depletion of predominantly pyrimidine nucleotides was observed, COX14[M19I] liver samples showed an enrichment of di- and tri-nucleotides (Fig. 6, Supplementary Fig. 7b). Increased levels of orotate and carbamoyl aspartate in COX14[M19I] liver samples indicate enhanced pyrimidine synthesis. Decreased levels of metabolites of the oxidative and non-oxidative phase of the pentose phosphate pathway, which delivers ribose and reduction equivalents for purine biosynthesis, could indicate increased usage of these metabolites for nucleotide biosynthesis. Moreover, we found increased levels of citric acid cycle metabolites in COX14[M19I] samples, which could reflect a boosted catabolic metabolism (Fig. 6, Supplementary Fig. 7b). Accordingly, nucleotide deficiency did not contribute to the release of mitochondrial nucleic acids in COX14[M19I] mice liver.

To assess the extent of damage mitochondria exhibited in the COX14[M19I] mice, primary hepatocytes were cultured and examined for different mitochondrial health parameters. The mitochondrial membrane potential in COX14[M19I] samples was significantly lower than that of wild type controls (Fig. 7a). Interestingly, COX14[M19I] hepatocytes elicited markedly increased mitochondrial ROS (Fig. 7b).

Since COX14[M19I] livers demonstrated an increase in SERBP-mediated cholesterol biosynthesis pathway as well as mitochondrial ROS production, we hypothesized that either of the two could trigger mitochondrial damage and subsequent nucleic acid release. To test this directly, we used respective inhibitors, Fatostatin and NAC, to assess if they inhibited type I IFN mediated inflammation. Additionally, known inhibitors of cytosolic nucleic acid sensing pathways were included in the analysis (Supplementary Fig. 8a). The treatment with Fatostatin did not alter the expression of ISGs, demonstrating that SERBP-mediated cholesterol biosynthesis did not contribute to the inflammatory response (Supplementary Fig. 8b). In contrast, NAC treatment caused the strongest reduction in ISGs expression amongst all tested reagents (Fig. 7c, Supplementary Fig. 8c–f). Furthermore, treatment with NAC prevented the release of mtRNAs in hepatocytes and subsequent upregulation of ISGs at the protein level (Fig. 7d, e). Upon NAC treatment, the reduction in IFIT1 protein levels matched the observed *Ifit1* mRNA changes. For *Isg15*, we similarly observe a strong reduction. The protein levels of ISG15 further depend on the levels of USP18, which itself was reduced in the NAC treated sample. Unfortunately, due to unavailability of suitable specific primers, the mRNA levels of OAS1a could not be assessed. In addition, we could show that the NAC treatments reduced the mitochondrial superoxide levels (Fig. 7f). NAC treatment acts via cellular glutathione and thereby quenches ROS via the glutathione pool. Therefore, in a complementary approach we utilized Mito-TEMPO, which acts on superoxide as a superoxide dismutase mimetic. Similar results as seen for NAC were obtained with Mito-TEMPO (Fig. 7g and Supplementary Fig. 8g, h). Accordingly, increased mitochondrial ROS production was causal for the inflammatory phenotype observed in COX14[M19I] livers.

In conclusion, instability of the COX14[M19I] protein and concomitant defects in complex IV assembly and function led to a

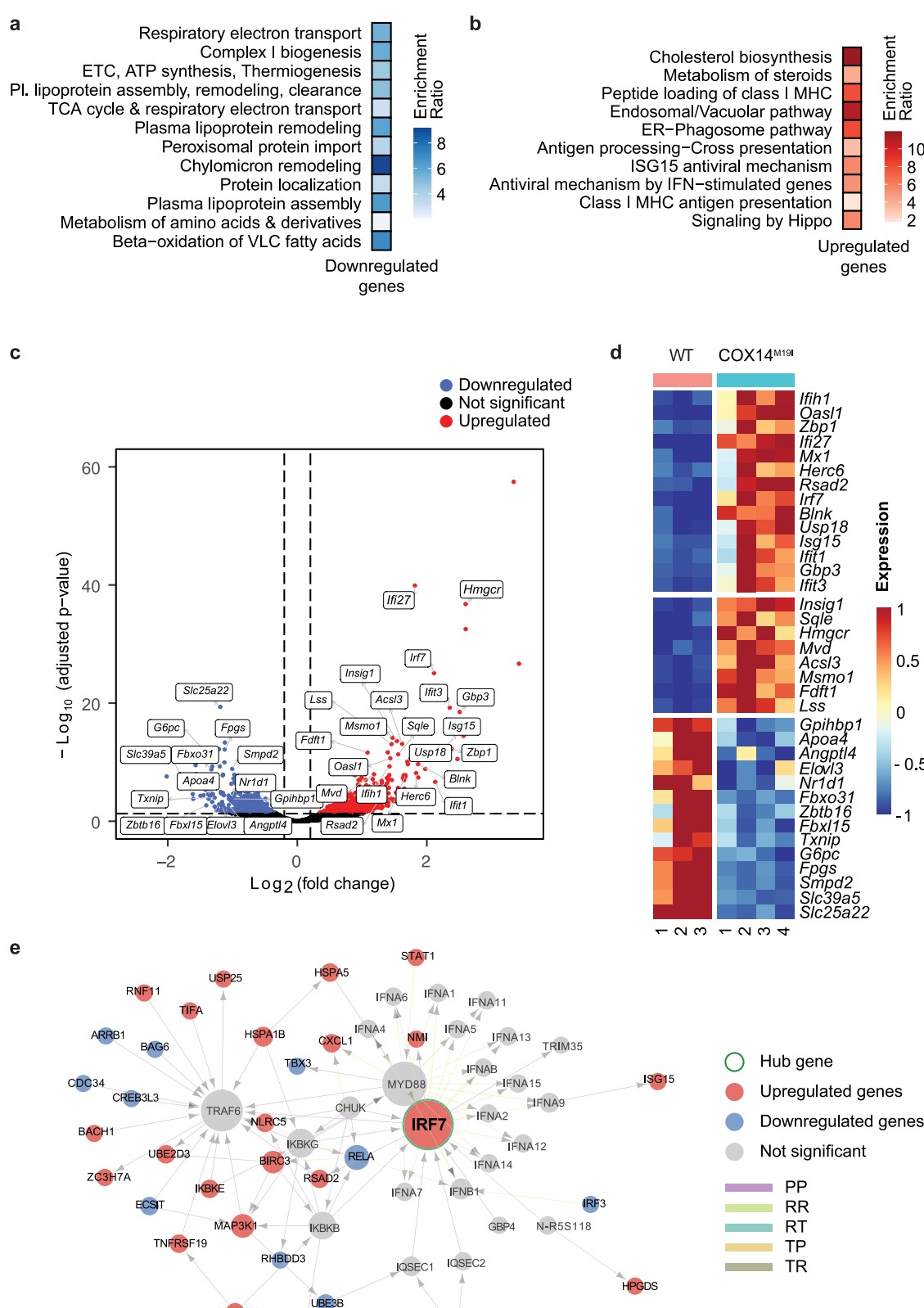

perturbation of mitochondrial oxidative phosphorylation with increase in mitochondrial ROS production. The damage inflicted by ROS lead to mitochondrial damage and subsequent nucleic acid release. Sensing of the "foreign" mitochondrial RNAs in the cytoplasm instigated the inflammation and subsequent liver pathology.

## Discussion

Mice models are important assets to study the molecular pathology of OXPHOS-dysfunction in tissues. Here we assessed how defects in mitochondrial translation in a patient-resembling mouse model, COX14^M19I, affect cellular health and crosstalk of metabolic pathways in different tissues. Amongst the analyzed tissues, liver was the most

**Fig. 3 | Altered gene expression in COX14^MI9I mice. a** Over-representation analysis of genes significantly downregulated in COX14^MI9I samples and **b** over-representation analysis of genes significantly upregulated in COX14^MI9I samples. For both (**c**) and (**d**): Functional pathways were derived from the Reactome database, and statistically significant over-represented (enriched) pathways were determined with a cutoff of FDR < 0.05 (Benjamini–Hochberg procedure). Pathways are arranged in increasing order of FDR values from top to bottom, and the colour indicates the enrichment ratio. **c** Volcano plot presentation of differentially expressed genes between liver samples from WT and COX14^MI9I mice. *X*-axis denotes fold change in expression (log2 scale), *y*-axis denotes adjusted *p*-value (negative log10 scale) for the analysed genes in the data set (each dot represents a single gene). Blue and red dots represent genes significantly downregulated and upregulated respectively in the COX14 mutant samples compared to WT (*DESeq2;* cut-offs used: adjusted *p*-value < 0.05 and absolute value of fold change <1.15). Non-significant genes are depicted in black. Selected genes involved in pathways relevant for the study are labelled in the plot. **d** Heatmap representing normalized expression (centered and scaled) for genes labelled in **A** in WT and COX14^MI9I liver samples. **e** Irf7 interaction network, with first degree (direct) and second degree (neighbours of direct) connections of Irf7. (Green node: hub gene (Irf7); red nodes: genes upregulated genes in COX14^MI9I samples; blue nodes: genes upregulated genes in COX14^MI9I samples; grey nodes: interactors in the network which are not significantly up- or downregulated.) The different kinds of (directed) interactions considered were: protein-protein (PP), RNA-RNA (RR), transcription factor-protein (TP), transcription factor-RNA (TR) and RNA-transcription factor (RT) interactions. The size of the nodes denotes the degree (number of outgoing + incoming interactions).

affected. This pathology was already apparent at early age of the mice analyzed. It is interesting to note that despite the systemic defect in COX14 function, the impact on protein stability, COX1 synthesis and concomitantly amounts of complex IV differed significantly between tissues. The question of tissue specificity of mitochondrial disorders is a central and still unresolved topic of the field. It is likely that different metabolic demands, energy requirements thresholds for cellular damages contribute to tissue sensitivity. In case of COX14^MI9I, the liver and heart defects correlated with reduced complex IV activity. Brain and muscle displayed similar complex IV reduction, yet a brain phenotype was not apparent in our analyses. Moreover, as we notice the most drastic reduction of the mutant COX14 protein in liver, one may argue that the reduced levels of COX14 in liver correlate with the complex IV defect. Yet, this correlation is not as apparent in the other tissues. Accordingly, the molecular pathology of mitochondria seems to affect different tissues to different extent. It is difficult to directly compare the mice phenotype to the patient pathology, as the death of the patient occurred already 24 hours after birth. Yet, cardiomyopathy and hepatomegaly were reported tissue dysfunctions. While cardiac defects were also apparent in the COX14^MI9I mouse, information on inflammation and retinal state of the patient are not available.

The liver plays key roles in many metabolic processes to which mitochondria contribute[21]. Accordingly, defects in liver mitochondria are linked to impaired carbohydrate and fatty acid metabolism, which affect energy production and increase fat accumulation leading eventually to fatty liver diseases[22]. A metabolomic screen of COX14^MI9I mice liver samples showed a decrease of pentose phosphate pathway (PPP) metabolites and increased levels of TCA intermediates. Cellular redox homeostasis is linked with the PPP through generation of NADPH, which is also utilized for fatty acid synthesis. Furthermore, liver also derives NADPH from folate-mediated serine catabolism for lipogenesis[23]. Yet, both, serine and glycine, were found to be reduced in COX14^MI9I livers. Additionally, transcriptional upregulation of SREBP-mediated cholesterol biosynthesis genes was apparent that did not translate to changes in cellular lipid species. In the light of these observation, the tight interactions between mitochondria and lipid droplets remains unexplained.

Among transcripts that were upregulated in COX14^MI9I liver, we identified such mRNAs that are linked to IRF7-induced inflammatory signaling pathways. Typically, these type I IFN responses or expression of interferon-stimulated genes (ISGs) can be very crudely categorized into either positive or negative regulators of inflammation. Among these, the positive regulators are subdivided into a wide array of genes depending on the mode of inhibiting viral replication[24,25]. COX14^MI9I livers displayed significant expression of *Irf7*, *Ifit1*, *Ifit3*, and *Ifi27*, which function towards binding viral RNA. Another set of positive regulators *Gbp3* and *Oas1a*, which do not bind viral RNA rather restrict the GTP availability or promote RNAse activity respectively, were also found to be upregulated in mutant samples. However, the negative regulators *Usp18* and *Oasl1*, which act on inhibiting the activity of ISG15 and IRF7,

were also highly expressed. Typically, both the positive and negative responses are simultaneously active to limit the extent and duration of type I IFN responses[26]. Accordingly, COX14^MI9I mice seem to exhibit a balance between both the responses thus limiting host toxicity and morbidity. However, persistent type I IFN signaling leads to other detrimental effects such as development of autoimmune diseases and decreased pathogen clearance[27,28]. Interestingly, chronic exposure to IFNα also promotes T-cell differentiation, perturbs lipid metabolism and redox balance[29,30]. It affects the metabolic fitness of CD8^+ T-cells as well[31]. Thus, the immune phenotype observed in COX14^MI9I animals might be linked to the chronic type I IFN signaling from liver. Type I IFN signaling also affects the cholesterol flux in cells. It downregulates transcriptional activity of SREBPs[32,33]. ISGs like RSAD2 have been also reported to inhibit cholesterol biosynthesis[34]. Surprisingly, COX14^MI9I livers demonstrate the complete opposite phenotype. One possible explanation for this could be energetic insufficiency resulting from decreased complex IV activity.

Our analyses show that the Type I IFN-mediated inflammation in COX14^MI9I livers arose from RIG1-MDA5 based mtRNA sensing. mtRNA has been suggested to be released into the cytosol through herniation through BAK/BAX mediated pore formation[35,36]. mtDNA has been proposed to be released through VDAC oligomers[37], BAK/BAX macropores[38], or mitochondrial-derived vesicles[39]. Yet, the triggers for these release processes appear to be highly variable ranging from physical injuries, cellular stress, metabolite imbalance to defects of mitochondrial health and dynamics[19]. In case of COX14^MI9I liver mitochondria, increased mitochondrial ROS production triggered mtRNA release. Despite its importance as signaling molecule, ROS can affect mitochondrial structure by causing oxidative damage to the mitochondrial membrane with a concomitant decrease in integrity and function[40]. In isolated rat and mice mitochondria and MEF cells ROS was found to stimulate mtDNA release[37,41]. Furthermore, ROS has been shown to trigger the formation of mitochondrial-derived vesicles[42]. Here we find that the instability of the COX14^MI9I protein results in defective translation. The concomitant loss of complex IV leads to increased ROS production, mitochondrial damage, and mtRNA release especially in the liver (that displays the most severe complex IV phenotype), which induces type I IFN inflammation. Inflammatory patterns are also observed in other tissues, yet they appear to be not as pronounced as in case of the liver. Based on our observations on RNA release in liver tissue we concluded that a similar process is also in place in other tissues that display inflammatory responses. At the molecular level, our findings provide insights on how defective mitochondrial translation due to loss of COX14 function triggers ROS-induced type I IFN inflammation in liver leading to worsening pathology with time.

## Methods

### Animals

The guidelines from the German Animal Welfare Act and approved by the Landesamt für Verbraucherschutz und Lebensmittelsicherheit,

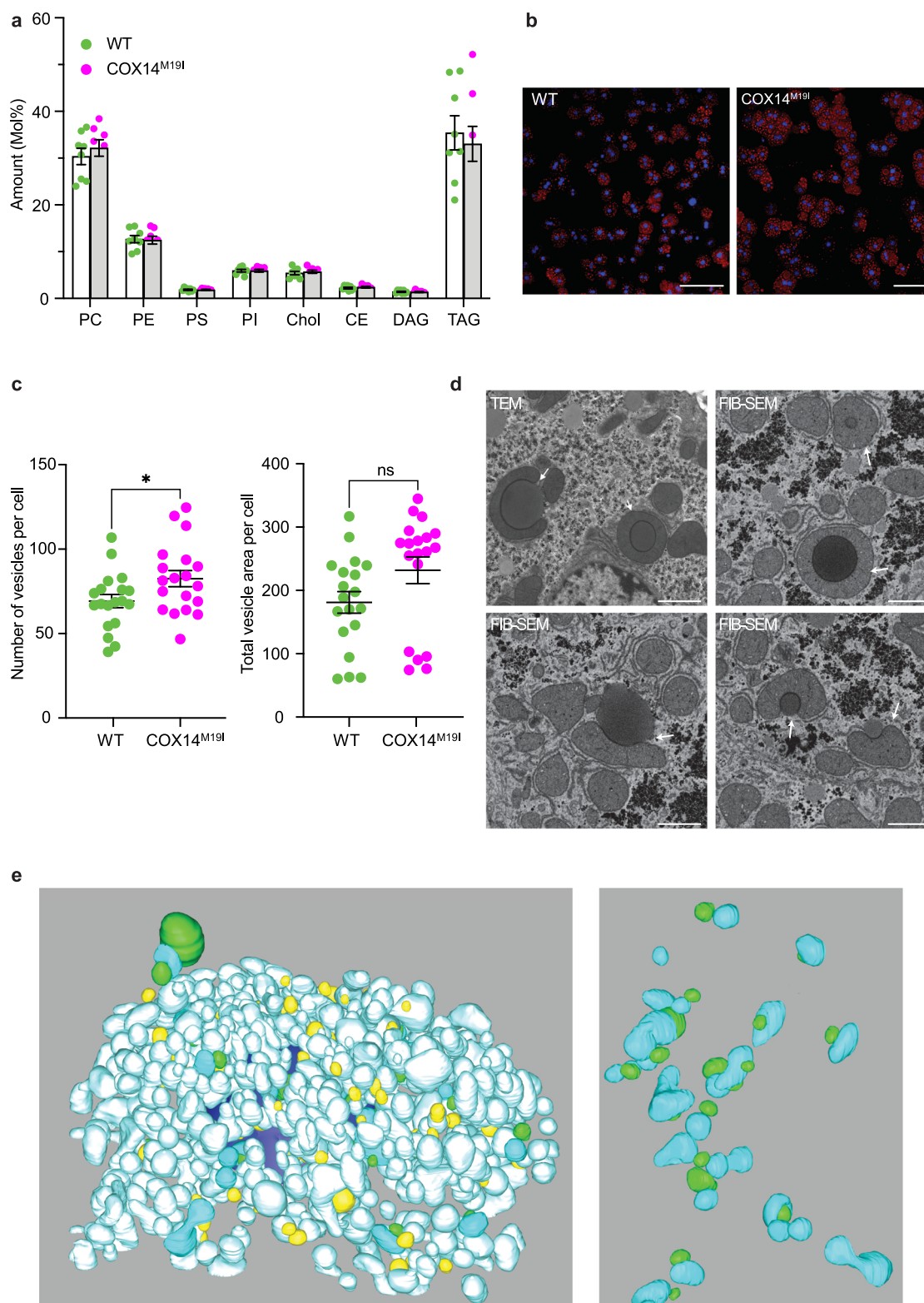

**Fig. 4 | Altered mitochondria lipid droplet contacts in COX14$^{M19I}$ mice. a** Total amount of different classes of lipid species in wild-type (WT) and COX14$^{M19I}$ mice liver samples analyzed by mass spectrometry. Means ± SEM, $n = 8$. **b** Cytochemistry of isolated primary hepatocytes from WT and COX14$^{M19I}$ mice using Oil Red O. $n = 3$, scale bar 100 μm. **c** Statistical analysis of images from **b**, number of lipid droplets per cell (left) and average lipid droplet area per cell (right). Means ± SEM, $n = 3$ biological replicates, > 6 technical replicates pre mouse, Unpaired $t$ test, left panel $p = 0.0374$; right panel $p = 0.0691$. **d** Representative TEM and FIB-SEM images of liver tissue samples from WT and COX14$^{M19I}$ mice. $n = 3$, scale bar 1 μm. **e** Representative 3D reconstructions of interaction between mitochondrion and lipid droplets observed in FIB-SEM images of liver tissue samples from COX14$^{M19I}$ mice. $n = 3$, scale bar 2 μm. Source data are provided as a Source Data file.

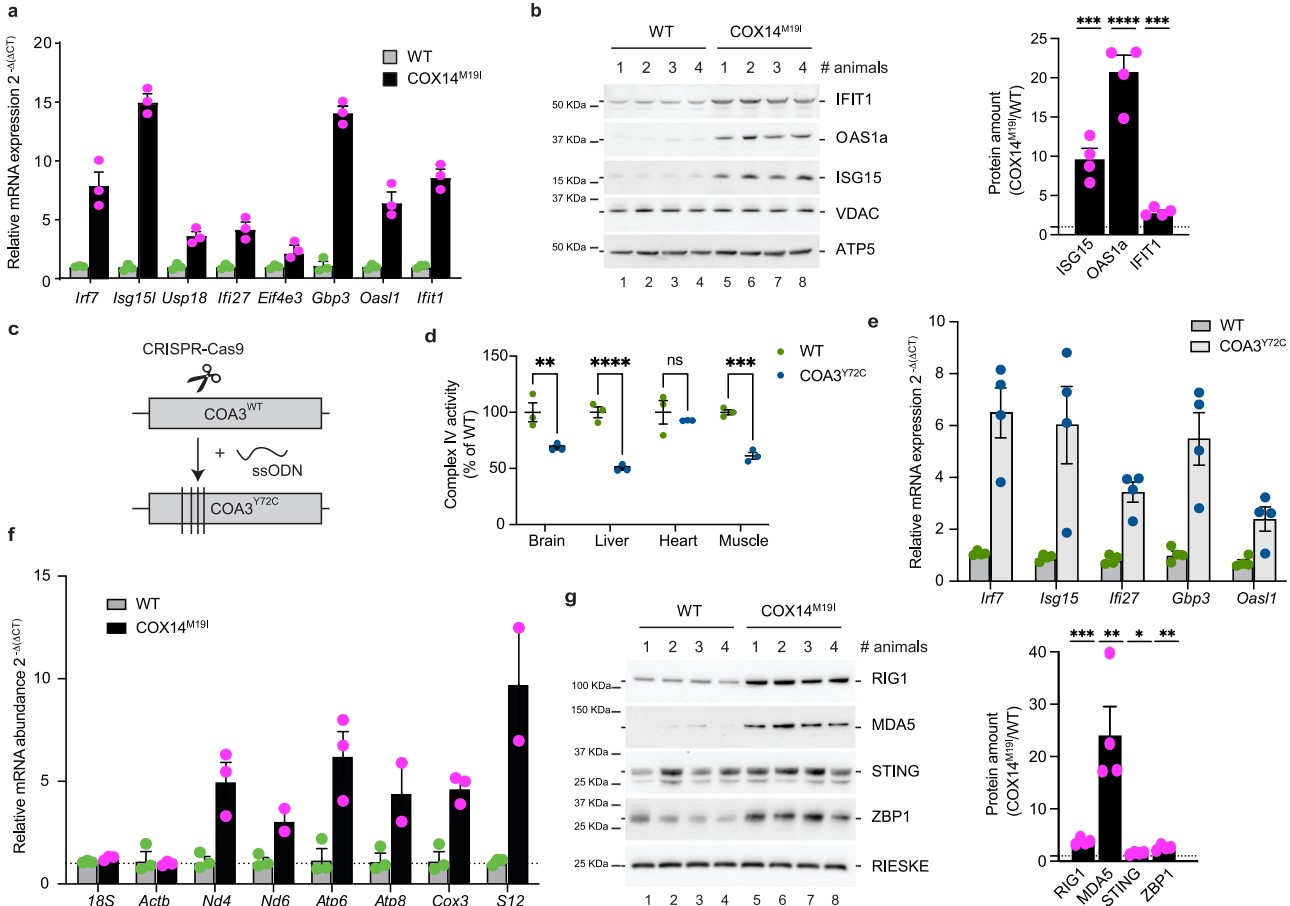

**Fig. 5 | Activation of inflammatory pathways in COX14^{M19I} and COA3^{Y72C} mice.**
**a** qPCR analysis of the gene expression of nucleic acid sensing pathway target genes in total mRNA isolated from 12-week-old WT and COX14^{M19I} mice liver samples. Means ± SEM, $n = 4$. **b** Western blot analysis and quantification of tissue lysates from 12-week-old WT and COX14^{M19I} mice livers with indicated antibodies. Means ± SEM, $n = 4$, Unpaired $t$ test, ISG15 $p = 0.0005$; OASL1a $p < 0.0001$; IFIT1 $p = 0.0003$. **c** Graphical presentation on the generation of the COA3^{Y72C} mouse line using CRISPR/CAS9-mediated double stranded cut in the *Coa3* allele and subsequent single stranded Oligonucleotide DNA (ssODN) mediated repair. **d** Enzyme activity of cytochrome *c* oxidase from 60-week-old WT and COA3^{Y72C} brain, heart, liver and

muscle plotted as percentage of average of WT. Means ± SEM, $n = 3$, One-way ANOVA, ns=non-significant, Brain $p = 0.0028$; Liver $p < 0.0001$; Muscle $p = 0.0004$ **e** qPCR analysis of gene expression of nucleic acid sensing pathway target genes in total mRNA isolated from 22-week-old WT and COA3^{Y72C} mice liver samples. Means ± SEM, $n = 4$. **f** Measurement of relative mRNA abundance in the cytosolic fractions of 12-week-old WT and COX14^{M19I} mice liver samples. Means ± SEM, $n = 4$. **g** Immunoblot analysis and quantification of tissue lysates from 24-week-old WT and COX14^{M19I} mice liver for indicated proteins. Means ± SEM, $n = 4$, Unpaired $t$ test, RIG1 $p = 0.0002$; MDA5 $p = 0.0051$; STING $p = 0.0416$; ZBP1 $p = 0.0015$. Source data are provided as a Source Data file.

---

Niedersachsen, Germany (AZ: 33.9-42502-04-14/1720) were followed for the maintenance of animals and performance of animal experiments. The animals were housed either in high barrier (SPF-specified pathogen free) areas in IVC (individually ventilated caging) on standard rodent chow to WT C57BL/6 N mice, with restricted access for animal care staff only. The COX14^{M19I} and COA3^{Y72C} mice were created using genome engineering based on the CRISPR/Cas9 method. To obtain zygotes for microinjection, donor female mice of the C57BL/6 N strain were super ovulated. Microinjection was then performed using a mix containing the CAS9 enzyme, an sgRNA targeting the either the COX14 or COA3 locus, and a single-stranded oligonucleotide DNA repair template, using micro manipulators. The resulting embryos were surgically transferred into pseudo-pregnant female mice of the NMRI or C57BL/6 N strain. Founders with edited genomes were identified through Sanger sequencing of genomic DNA from ear biopsies. These mice were crossed with C57BL6/6 N for two generations to ensure germline transmission and eliminate any possible mosaicism. Heterozygous animals with the same modification were subsequently mated to produce homozygous offspring. Finally, PACE (PCR Allele Competitive Extension) fluorescent endpoint genotyping was used for subsequent genotypings.

### Pronuclear injections for mice generation

The COX14^{M19I} mice (C57BL/6N-*Cox14^{em83Cecad}*), in which methionine 19 of the cDNA of Cox14 (Ensembl ENSMUST00000023761) is exchanged to isoleucine, was generated by pronuclear injection of CRISPR/Cas9 reagents (PMID: 29723268)[43]. 400 nM Alt-R™ guide RNA (5′-CACAGTGAGGAGCATCATCG −3′) targeting exon 2 of Cox14 and 500 nM standard desalted Ultramer™ single-stranded DNA repair oligonucleotides (5′- GCAGACCAACTCCAATCTGTCTT TGTAGGGGAA-GACATGCCATCTGCCAAGCAGCTAGCCGATATCGGCTACAAGACCTT CTCTGCATCGATAATGCTCCTCACTGTGTATGGGGGTTACCTCTGCA −3′) were injected with 200 nM Alt-R™ SpCas9 protein and 30 ng/μl SpCas9 mRNA (TriLink, L-6125-20) in C57BL/6 N zygotes. All components were purchased from Integrated DNA Technologies (USA) and animals obtained from Janvier Labs (France) if not stated otherwise. Genome editing for this line was performed at the in vivo Research Facility of the CECAD Research Center (University of Cologne, Germany) in compliance with the European, national and institutional guidelines and approved by the State Office of North Rhine-Westphalia, Department of Nature, Environment and Consumer Protection (LANUV NRW, Germany; animal study protocol AZ 84−02.04.2014.A372). Mice were anesthetized with ketamine (Ketaset,

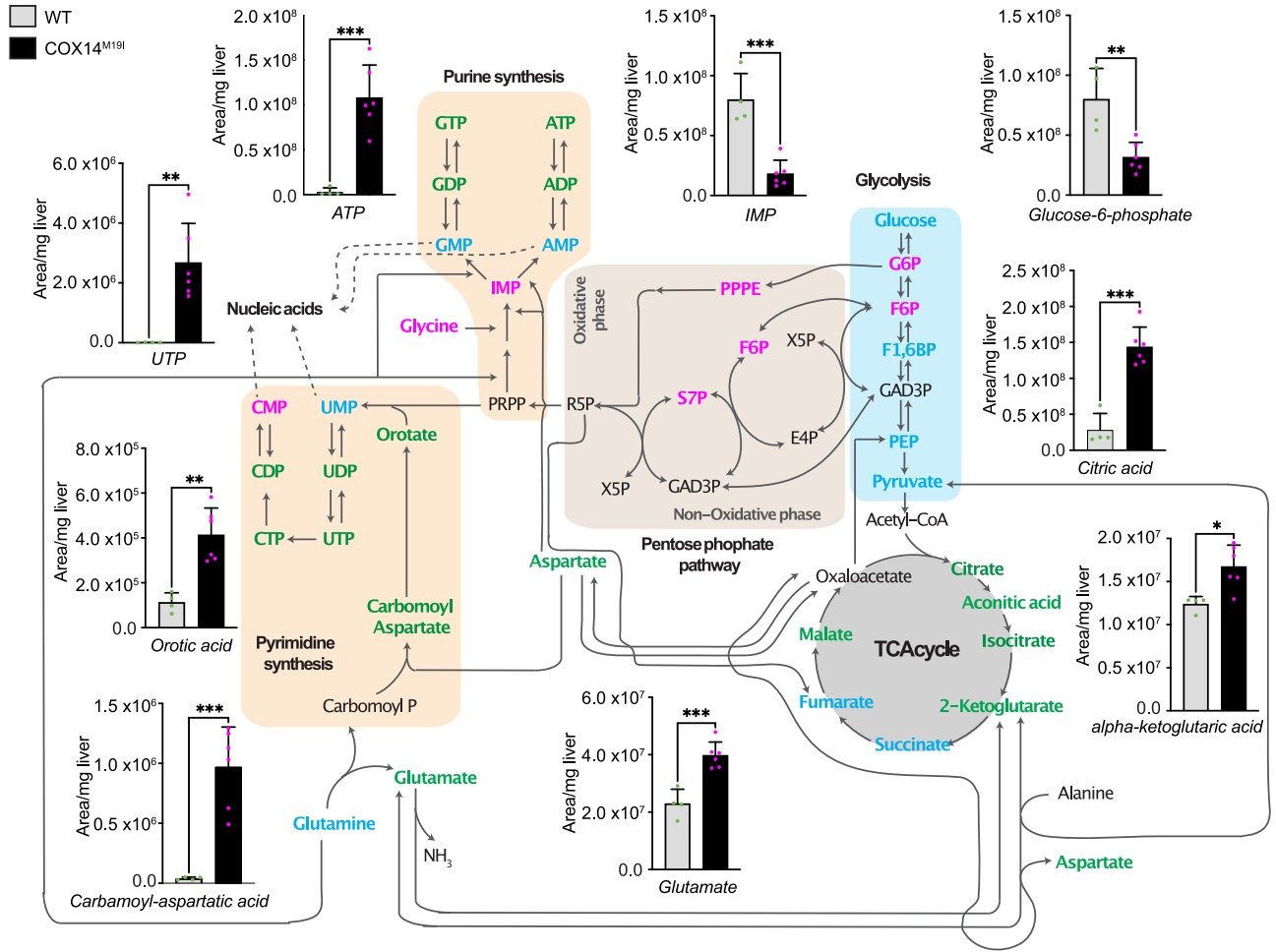

**Fig. 6 | Metabolic analysis of COX14$^{M19I}$.** Levels of different classes of metabolites determined by mass spectrometry in wild-type (WT) and COX14$^{M19I}$ mice liver samples. Means ± SD, n = 4 (WT); n = 6 (COX14$^{M19I}$), Unpaired t test, UTP p = 0.0039; ATP p = 0.0005; IMP p = 0.0003; Glucose-6-phosphate p = 0.0036; Citric acid p = 0.0001; Alpha-ketoglutaric acid p = 0.0107; Glutamate p = 0.0006; Carbamoyl-aspartic acid p = 0.0005; Orotic acid p = 0.0014. Blue−similar amount in WT and COX14$^{M19I}$; red−decreased amount in COX14$^{M19I}$ compared to WT; green−increased amount in COX14$^{M19I}$ compared to WT. PPPE, early metabolites of the oxidative phase of pentose phosphate pathway not individually resolved by the analysis. Source data are provided as a Source Data file.

Zoetis) and xylazine (Rompun, Bayer) and euthanized by cervical dislocation. Carprofen (Rimadyl, Zoetis) was used as analgesic after surgery. A similar strategy was used for the COA3$^{Y72C}$ mice, where the guide RNA (5′- CCTGAACCCACTCCGTCAGA −3′) targeted exon 2 of Coa3. The single-stranded DNA repair oligonucleotides (5′-AA GGGTGGCTAGACGAAAGGAATGCCCATGTGCTTTGTGGTTTCCTGAA CTCAACTTCTGTCAGGTGGGTATACCTTCTACTCGGTGGCCCAAGAG CGTTTCCTTGATGAGCTGGAAGATGA −3′) were used for repair.

### Mouse maintenance and handling
The mice are fed ad libitum with Ssniff chow V1534-300. (ssniff® R/M-H autoclavable Complete feed for rats and mice - maintenance, fortified). The mice are maintained at a 12:12 dark-light cycle with lights on at 4:30 am. The room temperature whetre the mice are kept is maintained at a constant temperature of 21 °C (+/−1 °C) and the humidity is about 50% (+/−5%). They are housed in a high level barrier (restricted entry; air-shower entry; gloves, mask, cap, change into sterilized scrubs and sterilized shoes; sterilization or disinfection of all supplies and equipment) in open cages under specific pathogen-free conditions. All mice of this husbandry unit are introduced into the unit via embryo transfer. The animal husbandry consists of five adjoining husbandry rooms. Hygienic monitoring by serological, microbiological and parasitological methods is performed according to the recommendations

of the FELASA. Parasitology is carried out on every strain present in the room Staphylococci (including Staphylococcus aureus) are found sporadically. The mice are sacrificaed via cervical dislocation.

### Isolation of primary hepatocytes and maintenance
Hepatocytes were isolated by a collagenase perfusion technique with modifications. Briefly, the mice were sacrificed by cervical dislocation and quickly dissected ventrally to expose the peritoneal cavity. The inferior vena cava was cannulated with a 24-gauge 3/4-inch angiocatheter (BD) and the portal vein was cut. The liver was perfused via the IVC with 100 mL of KR-EGTA Medium (120 mM NaCl, 4.8 mM KCl, 1.2 mM MgSO$_4$, 1.2 mM KH$_2$PO$_4$, 24 mM NaHCO$_3$, 0.25 mM EGTA, pH 7.4) at 37 °C, followed by perfusion with 70 mL of collagenase type IV (Worthington) in 1x PBS. After the liver was digested, it was dissected out and cut into small pieces and passed through a 100 mM strainer (BD). Hepatocytes were separated from nonparenchymal cells by low-speed centrifugation (100 g x 5 mins), and further purified by Percoll gradient centrifugation (50%, Sigma). The purified cells were washed once with Williams E Medium (supplemented with 10% FCS, 1% Gentamycin, 10$^{−8}$M Insulin). Cells were then plated in either Collagen or poly-L-Lysine coated plates or coverslips. The media was changed after 4 hours of plating to Maintenance Medium without FCS (William's E medium containing 1% Glutamax, 1% Non-Essential Amino Acids,

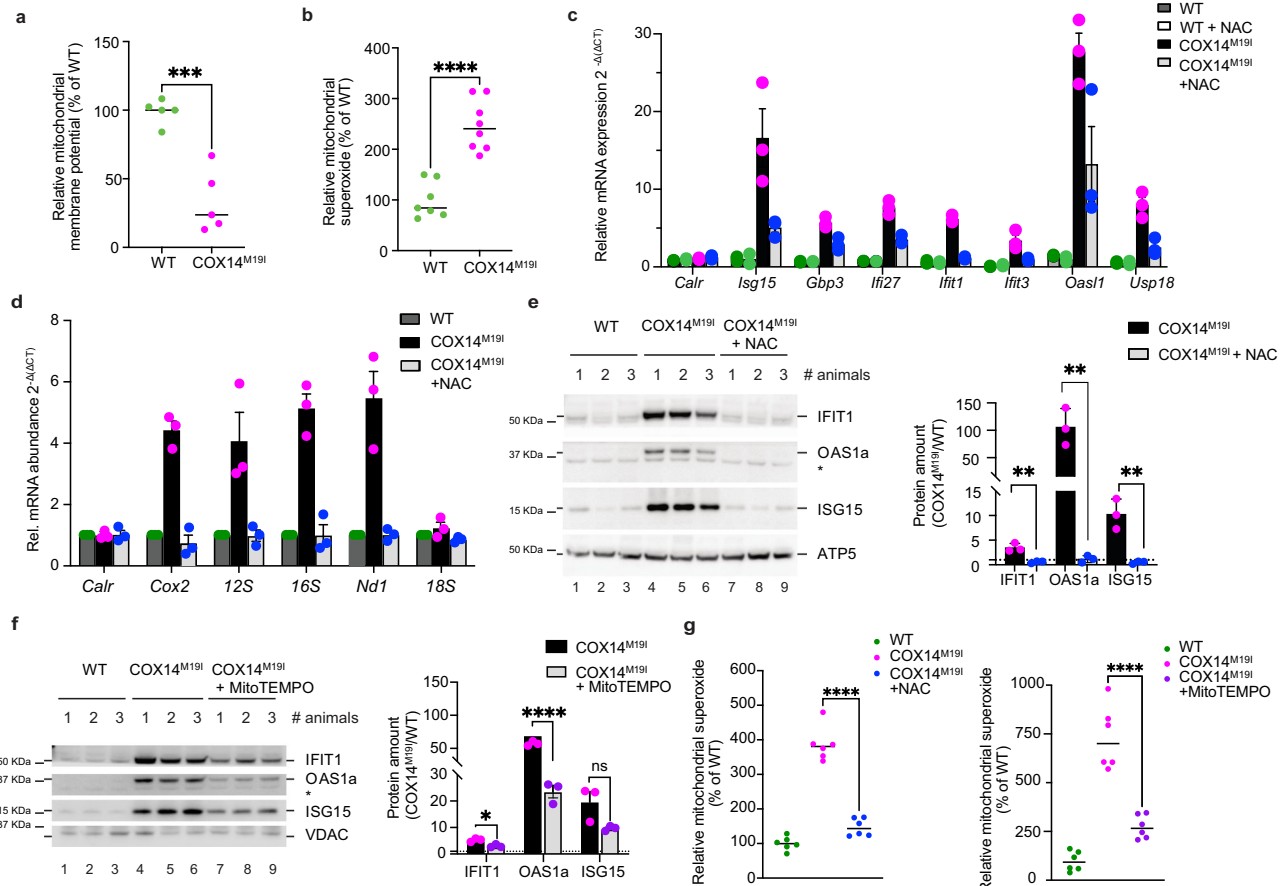

**Fig. 7 | mRNA release triggered by increased reactive oxygen species.**
**a** Mitochondrial membrane potential measured in WT and COX14^M19I primary
hepatocytes using TMRM staining. Means ± SEM, $n = 5$, Unpaired $t$ test, $p = 0.0003$.
**b** ROS production measured in WT and COX14^M19I primary hepatocytes using
MitoSOX Red Mitochondrial Superoxide Indicator. Means ± SEM, $n = 8$, Unpaired $t$
test, $p < 0.0001$. **c** qPCR analysis of gene expression of nucleic acid sensing
pathway target genes in total mRNA isolated from WT and COX14^M19I primary
hepatocytes treated with either vehicle control or NAC for 24 h. Means ± SEM, $n = 3$.

**d** Measurement of relative mRNA abundance in the cytosolic fractions from WT and
COX14^M19I hepatocyte fractionation samples. Means ± SEM, $n = 4$. **e, f** Western blot
analysis of cell lysates from WT and COX14^M19I primary hepatocytes with indicated
antibodies and quantification Means ± SEM, $n = 3$, Unpaired $t$ test, ns = non-sig-
nificant, **e**: IFIT1 $p = 0.0047$, OAS1a $p = 0.0061$, ISG15 $p = 0.0054$; **f** IFIT1 $p = 0.0113$,
OAS1a $p < 0.0001$. **g** ROS production measured in WT and COX14^M19I primary
hepatocytes using MitoSOX Red Mitochondrial Superoxide Indicator. Means ±
SEM, $n = 6$; $p < 0.0001$. Source data are provided as a Source Data file.

1% penicillin/streptomycin, 0.2% normocin (Invivogen), 2% B27
(GIBCO), 1% N2 supplement (GIBCO), 10 mM nicotinamide (Sigma-
Aldrich), 10 mM Y27632 (Peprotech), 1 mM A83-01 (Tocris), 3 mM
CHIR99021 (Peprotech), 25 ng/mL EGF (Peprotech), 50 ng/mL HGF
(Peprotech) and 100 ng/mL TNFa (Peprotech)). The cells were treated
with either 0.2 μM BX-795 (Sigma), 0.8 μM RU.521 (Sigma), 20 μM
Fatostatin (Sigma), 1 μM C-176 (Sigma), 50 μM BC-1215 (Hycultec), 0.1
μM GSK690693 (Sigma), 0.1μM MRT67307 (Sigma) or 10mM
N-acetylcysteine (NAC, Sigma) for 24 hours before harvesting them for
end point readouts.

### Mitochondrial isolation

**Mitochondrial isolation from mouse tissues.** Isolation of mitochon-
dria from mouse tissues (brain, heat liver and muscle) was performed:
Freshly isolated or snap frozen tissue samples were used for the iso-
lation of mitochondria. Heart and muscle samples were chopped into
small pieces and preincubated in Digestion buffer (1X PBS + 10 mM
EDTA + 0.05% Trypsin) for 30 min at room temperature. Mitochondrial
isolation from brain, heart, liver and muscle was done via mechanical
disruption at 16000 rpm using a Dounce homogenizer followed by
differential centrifugation. Mitochondrial Isolation Buffer_HM (67 mM
Sucrose + 50 mM Tris/Cl pH7.4 + 50 mM KCl + 10 mM EDTA + 0.5%
BSA; pH7.4) was used for the isolation of mitochondria from heart and
muscle. Mitochondria Isolation Buffer_L (10 mM Tris–MOPS, 1 mM

EGTA/Tris, 200 mM Sucrose; pH to 7.4) was used for the isolation of
mitochondria from liver. Mitochondria Isolation Buffer_B (75 mM
Sucrose, 215 mM D-Mannitol, 1 mM EDTA pH 7.4, 20 mM HEPES; pH to
7.4 (KOH)) was used for mitochondrial isolation from brain. The debris
was pelleted at 700 × g and the supernatant containing the mito-
chondria was separated. Pelleting of mitochondria was done at 8000 ×
g. Finally, the mitochondria were washed Mitochondrial Isolation Wash
Buffer (250 mM Sucrose + 3 mM EGTA/Tris + 10 mM Tris/Cl pH7.4;
pH7.4)[44].

**RNA isolation.** Cells and tissues were subjected to RNA isolation using
the TRIzol™ reagent, following the instructions provided by the
manufacturer (Thermo Fisher Scientific). The RNA obtained was either
used for reverse transcription into cDNA or subjected to RNA
sequencing.

**cDNA synthesis and qPCR.** First Strand cDNA Synthesis Kit (Thermo
Fisher Scientific) was used to transcribe isolated RNA using random
hexamer primers, resulting in the generation of cDNA. The cDNA was
then analyzed through qPCR using the SensiMix™ SYBR® Low-ROX kit
(Meridian Bioscience) with each sample pipetted in triplicate. The
reaction was carried out in a Quant studio 6 flex Cycler. The primer
sequences used for qPCR: *Cox14* (FP: ATGCCATCTGCCAAGCAGCTAG;
RP: CTGAGGTCTTCTGCTCTTCTGC), *Isg15* (FP: AAGCAGATTGCCCA

GAAGATTG; RP: CTGTACCACTAGCATCACTGTG), *s18* (FP: CTCAA-CACGGGAAACCTCAC; RP: CGCTCCACCAACTAAGAACG), *Ifi27* (FP: CAGTCTCTGGCACCATTCTA; RP: CAAAGCAGGTTCCTCTCCAATG), *Eif4e3* (FP: ACCACTTTGGGAGGAGGAGAG; RP: CACTCACGCT-GACTCCGATG), *Gbp3* (FP: TGGTGATGTAGAAAAGGGTGAC; RP: CTCACTGGAGTCCTTCACTT TG), *Oasl1* (FP: CAGCAGACCCCACCAA-CAAT; RP: GTGCTCTCTTCACCCTC CAG), *Ifit1* (FP: ATGGGAGA-GAATGCTGATGGTG; RP: AACTGGACCTGCTCTG AGATTC), *Usp18* (FP: TCGGTGATACCAAGGAACAGT; RP: ATCCTTCCAGGTGACCCAAC), *Actb* (FP: AGATGACCCAGATCATGTTTG; RP: GAGCATAGCCCTCGTA-GATG), *Nd4* (FP: ACCCGATGAGGGAACCAAAC; RP: AGCGTC-TAAGGTGTGTGTTGT), *Nd6* (FP: GGGTTTGGTGGATCGTTTTTAGG; RP: TCCCCAAGTCTCTGGATATTCCT), *Atp6* (FP: ACAGGCTTCCGA-CACAAACTA; RP: AGTAGCTCCTCCGATTAGGTGT), *Atp*8 (FP: CATCA-CAAACATTCCCACTGGC; RP: TGAGGCAAATAGATTTTCGTTCATT), *Cox3* (FP: TGCAGGATTCTTCTGAGCGT; RP: AGTAGTGGGACTTCTA-GAGGGTT), *s12* (FP: TGCAGGATTCTTCTGAGCGT; RP: AACTGCAAC-CAACCACCTTC), *Calr* (FP: AGTTCTTGGACGGAGATGCC; RP: GGCGGACAGTGCGTAAAATC), *s16* (FP: CGGCTAAACGAGGGTCCAAC; RP: TATTCTCCGAGGTCACCCCAA), *Nd1* (FP: TCCCCTACCAATACCA-CACCC; RP: GCTACGGCTCGTAAAGCTCC), *Cox2* (FP: AGACGAAAT-CAACAACCCCG; RP: GGCAGAACGACTCGGTTATCA), *Nd2* (FP: ATAGGGGCATGAGGAGGACT; RP: TGAGTAGAGTGAGGGATGGGTT), *Nd5* (FP: ACACATCTGTACCCACGCAT; RP: GGCGAGGCTTCCGAT-TACTA), *Irf7* (FP: TGAGCGAAGAGAGCGAAGAG; RP: GCCCACAGTAG ATCCAAGC).

**SDS PAGE and western blotting**. Standard methods were utilized to perform SDS PAGE and gradient Tricine-SDS PAGE (10–18%). Tissue and cell lysis were performed using T-PER™ Tissue Protein Extraction Reagent (Thermo Scientific) along with 1X Protease Inhibitor Cocktail and 1X PhosphoSTOP. The protein concentration was determined by utilizing a Bradford assay. For Western blotting, standard semi-dry transfer was employed. Antibodies used: Rabbit polyclonal Anti-IFIT1 antibody (ab111821) Abcam, dilution 1:100; Mouse monoclonal Anti-Oas1a Antibody (E-2): sc-365072 Santa Cruz, dilution: 1:1000; Mouse monoclonal Anti-ISG15 Antibody (F-9): sc-166755 Santa Cruz, dilution: 1:500; Rabbit polyclonal anti-VDAC Home-made antibody, dilution: 1:1000 Rabbit polyclonal anti-ATP5 Home-made antibody, dilution: 1:2500; Rabbit polyclonal anti-RIESKE Home-made antibody, dilution: 1:2000 Anti-Rig-I (D14G6) Rabbit mAb#3743 Cell Signaling Technologies, dilution: 1:1000; Anti- MDA-5 (D74E4) Rabbit mAb#5321 Cell Signaling Technologies, dilution 1:1000; Anti- STING (D2P2F) Rabbit mAb #13647 Cell Signaling Technologies, dilution 1:100; Mouse monoclonal ZBP1 Antibody (H-9): sc-271483 Santa Cruz, dilution 1:500; Rabbit polyclonal Anti-Lamin B1 antibody Abcam, dilution 1:500; Mouse monoclonal Anti-MTCO1 antibody [1D6E1A8] (ab14705) Abcam, dilution 1:1000;

**Blue native PAGE**. Cardiac mitochondria were solubilized in a mitochondria solubilization buffer (40 mM Tris/Cl pH 7.4, 100 mM NaCl, 20% Glycerol, 0.2 mM EDTA pH 7.4) containing 1% digitonin for 30 min at 4 °C, to a final concentration of 1 mg/ml. The solubilized samples were separated by 4–13% polyacrylamide gradient gel and transferred using a standard semi-dry method.

**In *organello* translation**. Mitochondria from freshly isolated liver tissue was used to perform in *organello* translation. 200 µg of isolated mitochondria were dissolved in freshly prepared 100 µl of Translation Buffer (25 mM sucrose, 100 mM Mannitol, 80 mM KCl, 5 mM MgCl$_2$, 1 mM Kpi pH7.4, 25 mM HEPES-KOH pH7.4, 6 mM ATP in H$_2$O pH7.2 KOH, 0.75 mM GTP I H$_2$O pH7,2 KOH, 12 mM Creatine phosphate, 10 mM Na-Succinate, 0.625 mg/ml Creatine Kinase, 9 mM 2-Ketoglutarate, 0.15 mM Amino Acid mix, 100 µg/ml Emetine, 100 µg /ml Cyclohexamide, 2.5 mM Malate, 1 mg/ml BSA, 1X Protease Inhibitor

Cocktail (PI)). The mitochondria were incubated at 37 °C for 10 min. 1 µCi $^{35}$S Methionine was added and the mitochondria were incubated for 1 hour at 37 °C while shaking. The reaction was stopped by adding 10 µM non-radioactive methionine. The mitochondria were pelleted down at 7000 xg, and washed once with Mitochondrial Isolation Wash Buffer. The samples were further used for steady state analysis.

To visualize the signal of translation products labeled with $^{35}$S Methionine, isolated mitochondria used for in *organello* translation, were loaded on a standard 10–18% Tris-tricine gradient gel and were further transferred on a PVDF membrane. The membrane was dried and was exposed for one day on a phosphor screen. The radioactive signal was visualized using Amersham Typhoon scanner (Amersham). The signal quantification was done using Image Quant LT (GE Healthcare).

**Proteomics**
**Sample preparation for LC-MS/MS.** Hepatocytes from three biological replicates of WT and of COX14$^{MI9I}$ were lysed in 8% sodium dodecyl sulfate (SDS), heated for 10 min at 99 °C and sonicated in Bioruptor (Diagenode) for 10 min using 30 s on/off cycles. Protein concentration in the lysate was determined using BCA assay kit (ThermoFisher Scientific) according to manufacturer's protocol. Proteins were reduced by incubation with 10 mM Dithiothreitol (DTT) for 30 min at 60 °C followed by alkylation with 40 mM iodoacetamide (IAA) for 30 min at 37 °C, which was quenched by addition of 20 mM DTT and incubated for 5 min at 37 °C. Protein clean-up was performed according to the procedure by Hughes et al.[45]. After washing, beads with bound sample were resolubilized in 50 mM TEA-B buffer. DNA and RNA were digested with Universal Nuclease (Pierce, 250 U/µl) for 30 min at 37 °C in presence of 1 mM MgCl$_2$. Proteins were digested with trypsin (MS grade, Promega) using 1:20 (w/w) trypsin-to-protein ratio and incubated overnight at 37 °C. Peptides were dried in SpeedVac concentrator (Eppendorf). and then re-dissolved in 50 mM TEA-B. Labelling reaction with respective TMT6plex labelling reagent (ThermoFisher Scientific) was conducted according to manufacturer's protocol. Labelled samples were pooled, dried in SpeedVac concentrator, and desalted using pre-packed C18-columns (Harvard apparatus). Desalted and dried peptides were dissolved in mobile phase A (10 mM NH$_4$OH, pH 10) and fractionated using reversed-phase chromatography under basic pH. Fractionation by bRP-HPLC was performed on an Agilent 1100 series HPLC system. Elution was performed at a flow rate of 60 µl/min using mobile phase A and B (10 mM NH$_4$OH in 80% ACN, pH 10) with a Waters XBridge C18 column (3.5-µm particles, 1.0 mM inner diameter, and 150 mM in length). The gradient was 5% B for 5 min, then changed to 10% B in 3 min, to 35% B in 34 min, to 50% B in 8 min, to 90% B in 1 min, to 95% in 5 min, back to 5% B in 2 min, then held at 5% B for 6 min (64 min total runtime). Flow through was collected for 6 min and then fractions of 60 µl were collected for the rest of the gradient. Fractions were combined into a total of 24 fractions plus the flow through, samples were dried in a SpeedVac concentrator, and stored at −20 °C for subsequent LC-MS/MS analysis. Labelling scheme for the comparison of cytoplasmic proteins in WT vs. COX14$^{MI9I}$ - (Sample, TMT label): 126, WT 1; 127, WT 2, 128, COX14$^{MI9I}$ 1, 129, COX14$^{MI9I}$ 2, 130, WT 3, 131, COX14$^{MI9I}$ 3. Isolated mitochondria from three biological replicates for each WT and COX14$^{MI9I}$ were treated with a similar protocol with the following changes. Lysis was performed with lysis buffer (4% SDS, 100 mM ethylenediaminetetraacetic acid (EDTA) Hepes pH 8.0, 2x complete protease inhibitor cocktail) instead of 8% SDS. Heating was performed at 95 °C for 5 min. Protein reduction and alkylation were performed with 10 mM Tris(2-chlorehyl)phosphine (TCEP) (Sigma-Aldrich) and 40 mM chloroacetamide (CAA) (Sigma-Aldrich) for 30 min at 55 °C. Proteins were precipitated with acetone (1:4 v/v sample-to-acetone ratio) overnight at −20 °C. The next day, the protein pellet was washed three times with ethanol and air-dried before solubilization with 1% Rapigest (Waters) in

50 mM TEA-B and incubation at 95 °C for 3 min. Then, four times the volume 100 mM TEA-B were added and proteins were digested with trypsin (1:20 w/w trypsin-to-protein ratio) overnight at 37 °C. After digestion, sample volume was reduced in the SpeedVac concentrator to ~15–20 µl. Samples were then subjected to TMT labelling and bRP-HPLC fractionation as described above. Labelling scheme for comparison of mitochondrial proteins in WT vs. COX14[M19I] – (Sample, TMT labelling): 126, WT 1; 127, COX14[M19I] 1; 128, WT 2; 129, COX14[M19I] 2; 130, WT 3; 131, COX14[M19I] 3.

**LC-MS/MS.** Dried peptides were solubilized in 2% (v/v) ACN 0.1% (v/v) TFA and injected onto a C18 PepMap100-trapping column (0.3 × 5 mM, 5 µm, Thermo Fisher Scientific) connected to an in-house packed C18 analytical column (75 µm x 300 mM; Reprosil-Pur 120 C18 AQ, 1.9 µm, Dr. Maisch GmbH). The columns were pre-equilibrated using a mixture of 95% buffer A (0.1% (v/v) formic acid in water), 5% buffer B (80% (v/v) ACN, 0.1% (v/v) FA in water). Liquid chromatography was operated by UltiMate 3000 RSLC nanosystem (Thermo Fisher Scientific). Peptides were eluted over 118 min using a gradient ranging from i) 5% to 8% buffer B over 4 min, ii) 8% to 32% buffer B over 72 min, iii) 32% to 50% over 30 min, and ending with a washing step at 90% of buffer B for 6 min and a re-equilibration step at 5% buffer B for 5 min. MS spectra were collected using an Orbitrap Fusion Tribrid mass spectrometer (Thermo Fisher Scientific). MS1 scans (350–1650 m/z) were acquired in positive ion mode with a resolution of 120,000 at 200 m/z, an automatic gain control (AGC) target of 250%, and with 50 ms maximum injection time. Precursor ions (allowed charge states 2–7, dynamic exclusion 40 s) were isolated using a 1.6 m/z isolation window and fragmented at normalized collision energy (HCD) of 30%. MS2 fragment spectra were acquired with a resolution of 30,000, AGC target of 500%, and 54 ms maximum injection time. The ten most intense fragment ions were selected for a subsequent synchronous precursor selection (SPS)-MS3 scan using an isolation window of 2 m/z and HCD collision energy of 55%. The SPS-MS3 spectra were acquired at a resolution of 30,000 and maximum injection time of 54 ms.

**LC-MS/MS data analysis.** Raw files were processed using MaxQuant version 2.1.4.0[46,47] using default settings except selection of reporter ions in MS3 (TMT6plex) for quantification. MS2 spectra were matched against canonical amino acid sequences of *mus musculus* proteins retrieved from Uniprot (August 2022). Differential expression analysis was conducted with Perseus software[46]. Corrected reporter intensity values were loaded into Perseus, contaminants and reverse hits were excluded, as well as protein hits with >70% missing values. Missing values were imputed from normal distribution with Perseus' default settings. Log2-transformed reporter ion intensities were then normalized using median polishing and subjected to statistical testing. Student's *T*-test was executed with S0 = 0, Benjamini-Hochberg FDR of 0.05. Differences in protein group intensities between COX14[M19I] mutant and wild type samples were expressed as log2 fold changes (COX14[M19I]/WT). Protein groups with *q*-value < 0.05 were considered as differentially expressed. Plots were generated with R programming language and R Studio (version 1.4.1106).

**Complex activity assays.** Isolated mitochondria from mouse hearts were utilized to perform Complex IV activity assays using the Complex IV Rodent Enzyme Activity Microplate Assay Kit (ab109911). ELISA-based plate assays were performed in accordance with the manufacturer's instructions (Abcam). To determine Complex IV activity, absorbance at 550 nm was measured. A Synergy H1 microplate reader (BioTek) was used to obtain the measurements.

**Real time respirometry**
**Mito stress test.** For the determination of the Oxygen Consumption rate (OCR) in isolated mitochondria a XF96e Extracellular Flux Analyzer was used. To measure basal respiration in isolated mitochondria, a Mitochondrial Assay Buffer was used. The buffer contained 70 mM Sucrose, 210 mM Mannitol, 5 mM HEPES, 1 mM EGTA, 10 mM $KH_2PO_4$, 5 mM $MgCl_2$, 0.5% BSA, and 40 mM ADP (added freshly). Two different supplements were used: either 10 mM Pyruvate, 4 mM Malate, and 4 mM ADP, or 10 mM Succinate and 4 mM ADP. Oxygen consumption was measured under different metabolic conditions by adding 3 µM Oligomycin, 4 µM CCCP, and 0.5 µM Antimycin/Rotenone.

**Palmitate oxidation stress test.** For the purpose of measuring long chain fatty acid oxidation in isolated mitochondria from mouse liver, a XF96e Extracellular Flux Analyzer was used. The mitochondria were freshly isolated at 4 °C as described above and plated in a Seahorse plate resuspended in Respiration Buffer (214 mM Sucrose, 12.8 mM KCl, 0.86 mM EGTA, 4.3 mM $MgCl_2$, 25.7 mM $K_2HPO_4$ 80µM Palmitoyl, 4.3µM Carnitine, 45 mM Malate, 3.7 mM ADP; ph 7.4) at room temperature. Addition of 0.2 mM ETO in the buffer was used as negative control. The basal Oxygen consumption rate was measured.

**Serum biochemistry.** Blood samples were taken from isoflurane-anesthetized mice by retro-bulbar sinus puncture with non-heparinized glass capillaries. Samples were collected in two different types of coated tubes (Lithium/Heparin (KABE cat # 078028) and EDTA (KABE cat # 078035). After collection each sample was mixed by gentle inversion and then kept on RT until centrifugation. Samples were centrifuged within 1 hour of collection at 5000 × g, for 10 min at 8 °C. Plasma is analyzed on Beckman AU480 biochemical analyzer. All calibration and quality control samples are within expiration period and measured values fulfilled required parameters for each method used according to the IMPC (International Mouse Phenotyping Consortium) core pipeline (https://www.mousephenotype.org/impress/index).

**Immunophenotyping.** The methodology was based on standard immunophenotyping protocols of the Adult and Embryonic Phenotype Pipeline that has been agreed by the research institutions involved MPReSS -International Mouse Phenotyping Resource of Standardised Screens (https://www.mousephenotype.org/impress/protocol/174/7). According to these guidelines, we use mouse spleenocytes for standard immunophenotyping.

**Histology.** Tissues were fixed in 4% paraformaldehyde, processed on a Leica ASP6025 automatic vacuum tissue processor, and embedded in a Leica EG1150 H + C embedding station. Sections (2 µm) were obtained using a Leica RM2255 rotary microtome, then deparaffinized and hydrated in xylen and descending ethanol solutions before being stained with Picrosirius red (Direct red 80) (Sigma Aldrich) using 0.5 g picrosirius powder stain per 500 mL picric acid solution. Weigert's haematoxylin kit (Sigma Aldrich) was used to counterstain the nuclei, following the manufacturer's instructions. Stained slides were dehydrated, mounted with coverslips in an automatic staining system (Leica ST5010-CV5030), and digitalized in a slide scanner AxioScan Z.1 (Zeiss).

**OCT and electroretinography**
**Optical coherence tomography (OCT).** In order to visualize the retinal morphology, the spectral-domain optical coherent tomography imaging system adapted to mouse eye (Heidelberg Engineering, Germany) was used. Examined mice were anaesthetized similarly as is described for Electroretinography. The eyes were dilated by a drop of 0.5% Atropine (Ursapharm, Prague, Czech Republic). Transparent eye gel Vidisic (Vidisic, Bausch&Lomb, Prague, Czech Republic) was used to prevent eyes from drying. The eyes were fitted with +100 dioptric lenses (Roland Consult, Brandenburg, Germany) and the mice underwent an imaging session[48].

Retinal thickness was assessed by an automatic segmentation algorithm complemented with minor manual corrections. Animals were scanned repeatedly at the age of 5, 15 and 30 weeks.

**Electroretinography.** Full-field single-flash electroretinography (ERG) was recorded under general anesthesia (i.m. Tiletamine + Zolazepam, 30 + 30 mg/kg, supplemented with 3.2 mg/kg Xylazine) using RETIanimal system (Roland Consult, Brandenburg, Germany). Animals were placed on a heated pad (38 °C), their pupils dilated by a drop of 0.5% Atropine (Ursapharm, Prague, Czech Republic) and eyes protected against drying by transparent eye gel (Vidisic, Bausch&Lomb, Prague, Czech Republic). Golden wire rings placed on the cornea served as active electrodes, reference subdermal electrode was in the central line near the snout and another subdermal electrode near the tail base represented the ground electrode. Scotopic ERG: mice were dark-adapted overnight; stimuli were single flashes of white light of luminances distributed in 9 exponential steps between 0.001 and 10 cd.s.m − 2. Photopic ERG: mice were exposed to 30 cd.m − 2 white light for 2 min. and recording continued with this background illumination, stimuli were single flashes between 1 and 100 cd.s.m − 2 in 5 intensity steps. For each stimulation intensity, 20 responses were collected and averaged and the amplitude of a, b, and c- wave, respectively, was measured on the averaged signal[48]. Experiments were approved by the Animal Care and Use Committee of the Czech Academy of Sciences.

**Echocardiography.** In order to visualize the small mouse heart and record the rapid heart rates, the cardiac ultrasound imaging is acquired using the Vevo 2100 Imaging System (VisualSonics, Inc.) with a 30 MHz transducer (MS400) operating at a frequency that provides highly reliable and reproducible image quality. Echocardiography is performed on anaesthetized (isoflurane) mice and during imaging, the concentration of anesthesia is controlled to maintain a heart rate of 450–500 beats/min. The primary echocardiographic screen includes two-dimensional (2D) B-mode ("brightness" mode) imaging view of left ventricle (LV) along the parasternal long axis (PLAX) and M-mode ("motion" mode) image of the heart in short-axis view (SAX) for accurate linear measurements of left ventricular internal dimensions.

A proper cardiac B-mode view allows visualization of the left atrium and ventricle, a small portion of the right ventricular wall and the output of aorta. The apex and the beginning of ascending aorta of the heart lay on the same horizontal line. M-mode echocardiography was performed at the ventricular level at the papillary muscle level and leads to a 1D high resolution temporal course of the diameter changes of the LV and the wall thickness of the anterior and posterior LV wall in systole as well in diastole. All M-mode echocardiography measurements follow the guidelines of the American Society of Echocardiography. The SAX M-mode images were used to measure left ventricular end-diastolic internal diameter (LVEDD), left ventricular end-systolic internal diameter (LVESD), and diastolic and systolic posterior wall thickness (LVPW) in three consecutive beats. Fractional shortening (FS) was calculated as FS% = [(LVEDD-LVESD)/LVEDD]x100. Ejection fraction (EF) was calculated as EF% = 100 x ((LVvolD-LVvolS)LVvolD) with LVvol = ((7.0/(2.4 + LVID) x LVID3). The corrected left ventricular mass (LV Mass Cor) was calculated as LV Mass Cor=0.8 (1.053 x ((LVIDD + LVPWD + IVSD)3 - LVIDD3)). The Stroke volume (SV) was the volume of blood pumped from one ventricle of the heart with each beat. The Stroke volume of the LV was obtained by subtracting end-systolic volume (ESV) from end-diastolic volume (EDV). The cardiac output (CO) was the volume of blood pumped by the heart per minute. In addition, heart rate and respiratory rate were calculated by measuring three systolic intervals, respectively three respiratory intervals.

**RNA sequencing.** Prior to library preparation, RNA quality was assessed by measuring the RNA integrity number (RIN) using a Fragment Analyzer HS Total RNA Kit (Advanced Analytical Technologies, Inc.). Samples with a RIN over 8 were selected. For library preparation, 200 ng of total RNA was used to create non-stranded mRNA-Seq libraries with the TruSeq RNA Library Preparation Kit (#RS-122-2001, Illumina). The size range of the final cDNA libraries was determined using the SS NGS Fragment 1-6000 bp Kit on the Fragment Analyzer, yielding an average size of 340 bp. Accurate quantification of the final libraries was performed using the DeNovix DS-Series System. The libraries were sequenced on the Illumina HiSeq 4000 platform (SE; 1 × 50 bp; 30–35 million reads per sample).

**Data processing and bioinformatic analyses.** To process the sequencing data, an in-house software pipeline was used. Illumina's bcl2fastq (v2.20.0) converted the base calls in the per-cycle BCL files to the per-read FASTQ format from raw images and performed adaptor trimming and demultiplexing.

**Mapping and normalization.** Initially, sequences were aligned to the reference mouse genome using the RNA-seq alignment tool (version 2.7.8a) allowing for two mismatches within 50 bases. Subsequently, read counting was performed using featureCounts. Read counts were analyzed in the R/Bioconductor environment (version 4.0.5) using the DESeq2 package version 1.31.5. Candidate genes were filtered using an absolute log2-fold change >1 and FDR-corrected $P$ value < 0.05. Gene annotation was performed using mouse entries via biomaRt R package version 2.46.3.

Furthermore, FastQC (v 0.11.9) was used for quality control of the raw sequencing data. STAR (v2.7.3a)[49], was used to map the reads to the genome (mm10 for mouse data) with a maximum of two mismatches allowed per pair and default parameters. The featureCounts program from the Subread package (v2.0.0)[50,51] was used to count the number of aligned reads overlapping with exons for each gene.

To identify differentially expressed genes, the unwanted sources of variation were identified and corrected using the svaseq function from the sva R package (v3.40.0)[52]. The DESeq2 R package (v1.32.0)[53] was used to perform the differential expression analysis. Genes were considered significantly differentially expressed if they passed a cut-off of baseMean >= 50, fold change >1.15 or < −1.15, and adjusted $p$-value < 0.05. The up and downregulated genes were analyzed for overrepresentation (pathway enrichment) using the WebGestaltR R package (version 0.4.4)[54] with KEGG database terms[55].

Plots were created in R (version 4.1.0) using custom scripts. Overrepresentation (pathway enrichment) analysis of the up and downregulated genes was done using the WebGestaltR R package (version 0.4.4)[54]. The Irf regulatory network was built using a custom script using the specific hub gene and deregulated gene lists, by parsing the following databases for interactions involving the input genes: RegNetwork[56], STRING[57], RISE[58] and NPInter[59]. The different kinds of (directed) interactions considered were: protein-protein (PP), RNA-RNA (RR), transcription factor-protein (TP), transcription factor-RNA (TR) and RNA-transcription factor (RT) interactions. Cytoscape[60] was used to build customized networks and visualizations.

**Lipidomics.** Liver samples that were flash frozen underwent lipidomics analysis[19]: Lipid extractions were performed in the presence of internal lipid standards using an acidic liquid-liquid extraction (ABD) method using chloroform:methanol:HCl (20:40:0.3, vol:vol:vol)[61], except for acid-labile plasmalogens, which were extracted under neutral conditions[62]. As controls, blank extractions and extractions in the presence of internal lipid standard without samples were performed. As quality control measure, standards were titrated into complex lipid mixtures of known composition. To ensure that similar amounts of lipids were subjected to extractions, a test extraction (ABD) was performed to determine the concentration of PC as a bulk membrane lipid and to adapt extractions volumes to similar total lipid amounts. On

average, a total lipid amount of 3000 pmol was used for extractions. Lipid extractions were performed in the presence of internal lipid standards for each lipid class, with the standards resembling the structure of the endogenous lipid species. Following this approach, a relative quantification of lipid species was performed. Glass Pyrex tubes (16x100mM) and screw caps with PTFE-liner (Corning) were used for extractions. Glass pipettes and glass syringes were used for handling of samples in organic solvents. Lipid standards were added from stock solutions (1–10 μM, in chloroform) prior to extractions, using a master mix consisting of 50 pmol phosphatidylcholine (PC, 14:1/14:1, 20:1/20:1; 22:1/22:1, Avanti Polar Lipids), 50 pmol sphingomyelin (SM, d18:1/13:0, d18:1/17:0, d18:1/25:0, semi-synthesized[62], 100 pmol deuterated cholesterol ($D_7$-cholesterol, Avanti Polar Lipids), 28 pmol phosphatidylinositol (PI, 17:0/ 20:4, Avanti Polar Lipids), 25 pmol phosphatidylethanolamine (PE) and 25 pmol phosphatidylserine (PS) (both 14:1/14:1, 20:1/20:1, 22:1/22:1, semi-synthesized[62]), 25 pmol dia-cylglycerol (DG, 15:0/18:1-$D_7$, Larodan), 25 pmol cholesteryl ester (CE, 19:0, Avanti Polar Lipids, 9:0, 24:1, Sigma), and 24 pmol triacylglycerol (TG, D5-mixture, LM-6000, Avanti Polar Lipids), 5 pmol ceramide (Cer, d18:1 with N-acylated 14:0, 17:0, 25:0, semi-synthesized[62] or Cer d18:1/ 18:0-$D_3$, Matreya) and 5 pmol glucosylceramide (HexCer) (GlcCer d18:1/17:0, Avanti Polar Lipids), 5 pmol lactosylceramide (Hex2Cer, d18:1/17:0; Avanti Polar Lipids), 10 pmol phosphatidic acid (PA, 17:0/ 20:4, Avanti Polar Lipids), 10 pmol phosphatidylglycerol (PG, 14:1/14:1, 20:1/20:1, 22:1/22:1, semi-synthesized[62]) and 5 pmol lysopho-sphatidylcholine (LPC, 17:1, Avanti Polar Lipids). The phosphatidy-lethanolamine plasmalogen (PE P-) standard mix consisted of 22 pmol PE P-Mix 1 (16:0p/15:0-d3, 16:0p/19:0), 31 pmol PE P- Mix 2 (18:0p/15:0-d3, 18:0p/19:0), 43 pmol PE P-Mix 3 (18:1p/15:0-d3, 18:1p/19:0). Semi-synthesis of PE P- was performed using deuterated fatty acids[63]. The final chloroform phase was evaporated under a gentle stream of nitrogen at 37 °C. Samples were either directly subjected to mass spectrometric analysis, or were stored at −20 °C prior to analysis, which was typically done within 1–2 days after extraction. Lipid extracts were resuspended in 10 mM ammonium acetate in 60 μl methanol. Samples were analyzed on an QTRAP6500+ (Sciex) mass spectrometer, except for the measurement of cholesterol acetate, which was performed on a QTRAP5500 (Sciex), and TG and DG mea-surements, which were performed on a Q Exactive. Both MS systems were coupled to a robotic with chip-based (HD-D ESI Chip, Advion Biosciences) electrospray infusion and ionization via a Triversa Nanomate (Advion Biosciences). For QTRAP6500+ and 5500 mea-surements, 2 μl aliquots of the resuspended lipid extracts were diluted 1:10 in 10 mM ammonium acetate in methanol in 96-well plates (Eppendorf twin tec 96). For TG and DG measurements on a Q Exac-tive, 5 μl aliquots of the reconstituted lipid extracts were diluted with 15 μl 7.5 mM ammonium formate in isopropanol:methanol:chloroform (v:v:v, 4:2:1). For cholesterol determinations, the remaining lipid extract was again evaporated and subjected to acetylation[64] All MS settings and scan procedures are listed in Supplementary Data S1. Data evaluation was done using LipidView 1.3 beta (Sciex) for QTRAP 6500 + /5500 data or LipidXplorer 1.2.8.1 (https://zenodo.org/record/3570469[65,66]) and Peakstrainer[67] for Q Exactive data, and in-house-developed software (ShinyLipids and ShinyLipidcountr_alpha4). The amount for endogenous molecular lipid species was calculated based on the intensities of the internal standards.

**Oil O red staining.** Lipid bodies were stained using OilO according to the manufacturer protocol. Briefly, a stock solution of 10 mg/ml of OilO in isopropanol was thoroughly mixed and set aside shortly. 15 min later, a 3:2 V:V dilution was prepared in water and wet aside again for a few minutes. The diluted solution was filtered and used fresh for staining. Cells were fixed using 4% poly-formaldehyde solution in PBS for 30 min, then quenched for 15 min with a 100 mM $NH_4Cl_2$ solution in PBS. Following two quick washes with water and

one wash with 60% isopropanol, OilO was applied for 5 min. Several washes in water were followed and then the coverslips were fixed on glass slide using embedding medium containing Hoechst for nuclei staining.

Images were captured using an inverted Nikon Ti epifluorescence microscope (Nikon Corporation, Chiyoda, Tokyo, Japan) fitted with a Plan Apochromat 60×, 1.4 NA oil immersion objective, an HBO-100W Lamp, and an IXON X3897 Andor (Belfast, Northern Ireland, UK) camera. The NIS-Elements AR software (version 4.20; Nikon) was uti-lized to operate the camera. Prior to image analysis, the images were deconvolved as follows. First, the point spread function (PSF) was generated using the PSF generator Fiji plugin via Born and Wolf 3D optical model. Then, the DevonvolutionLab2 plugin from Fiji was applied using Richardson-Lucy algorithm. The deconvolved images were analyzed using a built-in macro in Fiji. First, a threshold was applied to the OilO staining channel via Otsu method and the particles with 0.5–1 circularity were selected. Then, the acquired mask was used to measure the area and the average fluorescent intensity of individual particles. Nuclei stainings were used to count the number of cells in each image.

**Electron microscopy.** For transmission electron microscopy mice at the age of 16 weeks were fixed by perfusion fixation[68] or by immersion fixation using 2.5% glutaraldehyde, 4% formaldehyde and 0.5% NaCl in 0.1 M phosphate puffer pH7.3 as fixing solution. After dissection of the liver, small pieces of tissue were processed following a modified rOTO protocol[69,70]. Briefly, samples were washed in 0.1 M phosphate buffer, incubated for 3 h at 4 °C in 2% $OsO_4$ (Science Services GmbH, Munich, Germany) and 0.25% $K_4[Fe(CN)_6]$ (Electron Microscopy Sciences), washed with $H_2O$ and incubated with 0.1% thiocarbohydrazide (Sigma-Aldrich) for 1 h at room temperature. For contrast enhancement the tissue was again with 2% $OsO_4$ for 90 min at room temperature. The samples were then washed with $H_2O$, contrasted overnight at 4 °C with 2% uranyl acetate (SPI-Chem) and washed again with $H_2O$, followed by dehydration in an increasing acetone series (30%, 50%, 75%, 90%, 3 × 100%). The tissue was infiltrated with increasing concentrations of Durcupan (Sigma-Aldrich, components A, B, C) in acetone. Fresh Durcupan with accelerator (component D) was added to the samples for 5 h before embedding the samples in resin blocks. The blocks were polymerized for 48 h at 60 °C. Ultrathin sections were obtained using a 35° diamond knife (Diatome AG, Biel, Switzerland) and an ultra-microtome (RMC PowerTome PT-PC, Science Services, Munich, Ger-many) and placed on 100 mesh hexagonal copper grids (Science Services GmbH, Munich, Germany). Electron micrographs were taken with an LEO912 transmission electron microscope (Carl Zeiss Micro-scopy Deutschland GmbH, Oberkochen, Germany) using a 2k on-axis CCD camera (TRS, Moorenweis, Germany).

For volume EM by focused ion beam-scanning electron micro-scopy (FIB-SEM) the blocks were trimmed with a 90° rectangular shaped trimming diamond knife (Diatome AG, Biel, Switzerland) and then attached to the SEM stub (Science Services, Munich, Germany) using silver filled epoxy (Epoxy Conductive Adhesive, EPO-TEK, Elec-tron Microscopy Service) and polymerized at 60° overnight. After coating the samples with a 10 nm gold layer with the sputter coater EM ACE600 (Leica Microsystems, Vienna, Austria) at 35 mA current the sample stubs were placed into the Crossbeam 540 FIB-SEM (Carl Zeiss Microscopy Deutschland GmbH). In the chamber, a protective 500 nm platinum layer was deposited on top of the region of interest using a 3 nA current. Atlas 3D (Atlas 5.1, Fibics, Canada) software was used to collect the 3D data. The block face was exposed with a 15 nA current, and a 7 nA current was used to polish the surface. The images were acquired at 1.5 kV with the ESB detector (450 V ESB grid, pixel size x/y 5 nm) in a continuous mill-and-acquire mode using a current of 1.5 nA for milling (z-step 10 nm). Image post-processing was performed using the image processing package Fiji[71]: Alignment of images was

performed using the SIFT algorithm. Then the images were cropped, the contrast inverted and smoothed using a Gaussian blur (sigma 1), and a local contrast enhancement was applied (CLAHE: blocksize 127, histogram bins 256, maximum slope 1.5).

**Metabolomics.** Samples were prepared and processed for LC-MS analysis of polar and lipid metabolites as detailed in Supplementary Methods. Anion-Exchange Chromatography Mass Spectrometry (AEX-MS) was used for the analysis of anionic metabolites. Whereas semi-targeted liquid chromatography-high-resolution mass spectrometry-based (LC-HRS-MS) analysis was performed for amine-containing metabolites.

**Samples extraction of polar metabolites for metabolomics.** For the metabolite extraction 20–30 mg of snap frozen liver tissue were collected in 2 mL Eppendorf tubes containing a 5 mM metal ball. These tubes were kept in liquid nitrogen before grinding them to a fine powder using a TissueLyser (Qiagen) set to 25 Hz for 1 min.

Immediately after homogenization 1 ml of −20 °C acetonitrile:methanol:water (4:4:2 (v:v:v)) mixture, containing 0.1 μL of U-$^{13}C^{15}N$ amino acid mix (Cambridge isotopes MSK_A2-1.2), each 0.1 μl of a 1 mg/mL stock solution of $^{13}C_{10}$ ATP, $^{15}N_5$ ADP and $^{13}C_{10}^{15}N_5$ AMP (Sigma) and 0.2 μL of a 100 μg/mL stock solution of deuterated citric acid (Sigma), was added. The extraction solvents were all (UPLC-grad, Biosolve). After addition of the extraction buffer, the samples were immediately vortexed for 10 seconds before incubating them for additional 30 min at 4 °C on an orbital shaker. To remove the insoluble material the metal balls were removed from each sample using a magnet and the tubes were centrifuged for 10 min at 21.000 × *g* at 4 °C. The cleared supernatant was transferred to fresh 1.5 mL Eppendorf tube and immediately concentrated to complete dryness in a speed vacuum concentrator at room temperature. Samples were either stored at −80 °C or processed immediately for LC-MS analysis.

**Anion-Exchange Chromatography Mass Spectrometry (AEX-MS) for the analysis of anionic metabolites.** Extracted metabolites were re-suspended in 400 μl of UPLC/MS grade water (Biosolve), of which 200 μl were transferred to p*olypropylene* autosampler vials (Chromatography Accessories Trott, Germany) before AEX-MS analysis.

The samples were analysed using a Dionex ionchromatography system (Integrion Thermo Fisher Scientific)[72]. Five μL of polar metabolite extract were injected in push partial mode, using an overfill factor of 1, onto a Dionex IonPac AS11-HC column (2 mM × 250 mM, 4 μm particle size, Thermo Fisher Scientific) equipped with a Dionex IonPac AG11-HC guard column (2 mM × 50 mM, 4 μm, Thermo Fisher Scientific). The column temperature was held at 30 °C, while the auto sampler was set to 6 °C. A potassium hydroxide gradient was generated using a potassium hydroxide cartridge (Eluent Generator, Thermo Scientific), which was supplied with deionized water (Millipore). The metabolite separation was carried at a flow rate of 380 μL/min, applying the following gradient conditions: 0–3 min, 10 mM KOH; 3–12 min, 10–50 mM KOH; 12–19 min, 50–100 mM KOH; 19–22 min, 100 mM KOH, 22–23 min, 100–10 mM KOH. The column was re-equilibrated at 10 mM for 3 min.

For the analysis of metabolic pool sizes the eluting compounds were detected in negative ion mode using full scan measurements in the mass range m/z 77–770 on a Q-Exactive HF high resolution MS (Thermo Fisher Scientific). The heated electrospray ionization (HESI) source settings of the mass spectrometer were: Spray voltage 3.2 kV, capillary temperature was set to 300 °C, sheath gas flow 50 AU, aux gas flow 20 AU at a temperature of 330 °C and a sweep gas glow of 2 AU. The S-lens was set to a value of 60.

The semi-targeted LC-MS data analysis was performed using the TraceFinder software (Version 4.1, Thermo Fisher Scientific). The identity of each compound was validated by authentic reference compounds, which were measured at the beginning and the end of the sequence. For data analysis the area of the deprotonated [M-H$^+$]$^{-1}$ or doubly deprotonated [M-2H]$^{-2}$ monoisotopologue mass peaks of every required compound were extracted and integrated using a mass accuracy <5 ppm and a retention time (RT) tolerance of <0.05 min as compared to the independently measured reference compounds. These areas were then normalized to the internal standards, which were added to the extraction buffer, followed by a normalization to the fresh weight of the analyzed sample.

**Semi-targeted liquid chromatography-high-resolution mass spectrometry-based (LC-HRS-MS) analysis of amine-containing metabolites.** The LC-HRMS analysis of amine-containing compounds was performed[73] using 50 μL of the available 400 μL of the above mentioned (AEX-MS) polar phase which were mixed with 25 μl of 100 mM sodium carbonate (Sigma), followed by the addition of 25 μl 2% [v/v] benzoylchloride (Sigma) in acetonitrile (UPC/MS-grade, Biosove, Valkenswaard, Netherlands). Derivatized samples were thoroughly mixed and kept at 20 °C until analysis.

For the LC-HRMS analysis, 1 μl of the derivatized sample was injected onto a 100 × 2.1 mM HSS T3 UPLC column (Waters). The flow rate was set to 400 μl/min using a binary buffer system consisting of buffer A (10 mM ammonium formate (Sigma), 0.15% [v/v] formic acid (Sigma) in UPC-MS-grade water (Biosove, Valkenswaard, Netherlands). Buffer B consisted of acetonitrile (IPC-MS grade, Biosove, Valkenswaard, Netherlands). The column temperature was set to 40 °C, while the LC gradient was: 0% B at 0 min, 0–15% B 0–4.1 min; 15–17% B 4.1–4.5 min; 17–55% B 4.5–11 min; 55–70% B 11–11.5 min, 70–100% B 11.5–13 min; B 100% 13–14 min; 100–0% B 14–14.1 min; 0% B 14.1–19 min; 0% B. The mass spectrometer (Q-Exactive Plus, Thermo Fisher Scientific) was operating in positive ionization mode recording the mass range m/z 100–1000. The heated ESI source settings of the mass spectrometer were: Spray voltage 3.5 kV, capillary temperature 300 °C, sheath gas flow 60 AU, aux gas flow 20 AU at 330 °C and the sweep gas was set to 2 AU. The RF-lens was set to a value of 60.

Semi-targeted data analysis for the samples was performed using the TraceFinder software (Version 4.1, Thermo Fisher Scientific). The identity of each compound was validated by authentic reference compounds, which were run before and after every sequence. Peak areas of [M + nBz + H]$^+$ ions were extracted using a mass accuracy (<5 ppm) and a retention time tolerance of <0.05 min. Areas of the cellular pool sizes were calculated according to the AEX-MS method.

**ROS and TMRM staining.** To measure mitochondrial membrane potential and superoxide anion production, Tetramethylrhodamin-methylester-perchlorat, TMRM and MitoSOX™ Red respectively were utilized to stain isolated hepatocytes at a concentration of 10μM and 3μM each, following the manufacturer's protocol (Invitrogen). BD-Canto flow cytometer (Becton Dickinson) was used to record 10,000 gated events per sample, which were analyzed using the FACS-Diva software to derive the final result.

**Software.** Data collection was done and analysis was preformed using softwares associated with the instruments used for the specific experiments, as described in the correcponding methods sections. In short; Western blot quantifications: ImageStudioLite Software (Version 5.2.5); Numerical data presentation and statistical analysis: Graph Pad Prism (Version 10.0.3); Proteomics: Perseus software and R Studio (version 1.4.1106); RNA sequencing: Illumina's bcl2fastq (v2.20.0), FastQC (v 0.11.9), STAR (v2.7.3a), R (version 4.1.0); Lipidomics: LipidView 1.3 beta (Sciex), LipidXplorer 1.2.8.1 (https://zenodo.org/record/357046923,24)

**Study design.** Sample size for each experiment was determined based on prior experience and established protocols within the field for that type of experiment. No data was excluded unless statistically

calculated for the specific experiment (see Source data). At least three biological replicates were done for each experiment. The exact number of biological replicates for each experiment is given in the Figure legends. Experiments were performed by two different researchers to assure randomisation. Male and female mice were used for all experiments and with comparable results. Both mutant and control groups were processed at the same time and using the same reagents. No blinding was used.

**Quantification and statistics.** Prism 9 for macOS software (GraphPad Software, San Diego, CA) was used to calculate the Standard Error mean (SEM) or Standard Deviation (SD) and assess the significance of differences between groups, unless otherwise specified. The tests employed were either unpaired non-parametric or paired non-parametric t-tests and ANOVA analysis, as appropriate. All replicates in the experiments presented are biological replicates unless stated otherwise in the figure legends.

### Reporting summary
Further information on research design is available in the Nature Portfolio Reporting Summary linked to this article.

## Data availability
The transcriptomics data generated in this study have been deposited in the GEO database under accession code GSE269535. The proteomics data generated in this study have been deposited in the PRIDE database under accession code: PXD041783. The Metabolomics data generated in this study have been deposited in the Zenodo database under accession code: doi (10.5281/zenodo.10640936) [https://zenodo.org/records/10640936]. The Lipidomics reporting checklists[74] generated in this study have been deposited in the Zenodo database under accession code: (https://doi.org/10.5281/zenodo.10891305), (https://doi.org/10.5281/zenodo.10891308). The Lipidomics MS raw data generated in this study has been deposited in the Metabolights database under accession code (MTBLS9823) [https://www.ebi.ac.uk/metabolights/MTBLS9823]. Source data are provided with this paper.

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

## Acknowledgements

We are grateful to Dr. Salinas for discussion. Supported by the Deutsche Forschungsgemeinschaft under Germany's Excellence Strategy EXC 2067/1-390729940; SFB1002 (A06, PR; D04, A.F.), SFB1286 (A6/A10, PR & HU; B06, A.F.), FOR2848 (P04, PR/SJ; P08, M.O.) and the Max Planck Society (P.R.; S.J.). B.B. gratefully acknowledge the data storage service SDS@hd supported by the Ministry of Science, Research, and the Arts Baden-Württemberg (MWK) and the DFG through grant INST 35/1314-1 FUGG and INST 35/1503-1 FUGG. Experiments and analysis performed at the Czech Centre for Phenogenomics at the Institute of Molecular Genetics were supported by the Czech Academy of Sciences RVO 68378050 and by the project LM2023036 provided by the Ministry of Education, Youth and Sports (MEYS) of the Czech Republic and by the project CZ.02.1.01/0.0/0.0/18_046/0015861 by MEYS and ESIF. RNA sequencing was carried out at the NGS- Integrative Genomics Core Unit of the University Medical Center Goettingen.

## Author contributions

A.A., A.B., S.W., T.G., K.U.F., R.Y., A.C., D.D., A.M., S.K., I.S., Z.N., M.P., M.R., G.K., S.E.T., D.R., S.M., P.S., C.L., and P.G. performed experiments. A.A., A.B., S.W., D.D. and A.M. prepared the figures. P.R., A.A., J.P., R.S., W.M., B.Z., P.S., H.U., A.F., B.B. and S.J. designed the study and/or provided supervision. P.R., A.A. and A.B. wrote the manuscript.

## Funding

## Competing interests

A.C. is now an employee of Dewpoint Therapeutics GmbH. C.L. is now an employee of BioNTech. The other authors declare no competing interests.

## Additional information

[1]Department of Cellular Biochemistry, University Medical Center Göttingen, 37073 Göttingen, Germany. [2]Cluster of Excellence "Multiscale Bioimaging: from Molecular Machines to Networks of Excitable Cells" (MBExC), University of Göttingen, Göttingen, Germany. [3]Department of Molecular Biochemistry, University Medical Center Göttingen, 37073 Göttingen, Germany. [4]Department of Psychiatry and Psychotherapy, University Medical Center Göttingen, Göttingen, Germany. [5]Department for Epigenetics and Systems Medicine in Neurodegenerative Diseases, German Center for Neurodegenerative Diseases (DZNE), Göttingen, Germany. [6]Bioanalytical Mass Spectrometry Group, Max Planck Institute for Multidisciplinary Sciences, Göttingen, Germany. [7]Institute for Clinical Chemistry, University Medical Center Göttingen, Göttingen, Germany. [8]Czech Centre for Phenogenomics, Institute of Molecular Genetics of the CAS, v.v.i, 252 50, Vestec, Czech Republic. [9]Cluster of Excellence Cellular Stress Responses in Aging-associated Diseases (CECAD), Faculty of Medicine and University Hospital Cologne, University of Cologne, Cologne, Germany. [10]Laboratory for Electron Microscopy, Max Planck Institute for Multidisciplinary Sciences, Göttingen 37077, Germany. [11]Max Planck Institute for Multidisciplinary Science, Department of Neurogenetics, 37077 Göttingen, Germany. [12]Institute of Pathology, University Medical Center Göttingen, Göttingen, Germany. [13]Heidelberg University Biochemistry Center (BZH), 69120 Heidelberg, Germany. [14]Max Planck Institute for Biology of Ageing, 50931 Köln, Germany. [15]German Center for Cardiovascular Research (DZHK), partner site Göttingen, Göttingen, Germany. [16]Department of NanoBiophotonics, Max Planck Institute for Multidisciplinary Sciences, 37077 Göttingen, Germany. [17]Clinic of Neurology, University Medical Center Göttingen, 37075 Göttingen, Germany. [18]Fraunhofer Institute for Translational Medicine and Pharmacology ITMP, Translational Neuroinflammation and Automated Microscopy, Goettingen, Germany. [19]Max Planck Institute for Multidisciplinary Sciences, D-37077 Goettingen, Germany. ✉e-mail: peter.rehling@medizin.uni-goettingen.de

