## [Peer Review File · Nature Communications]

REVIEWER COMMENTS

Reviewer #1 (Remarks to the Author):

The assembly of cytochrome c oxidase (COX), the last enzyme of the mitochondrial respiratory chain, requires the function of numerous assembly factors, two of which, COX14 and COA3, are required for the expression and assembly of COX subunit 1. Mutations in both factors have been reported in patients with mitochondrial disorders, although with different severity. How mutations comparably affecting COX lead to different pathological manifestations remains to be fully elucidated. To gain insights into the pathophysiology of these disorders, in the manuscript by Aich et al, the authors generated and characterized two murine models recapitulating the COX14 and COA3 patient mutations. The mutation in COX14 is associated with multisystemic alterations in several tissue, with the liver being the most affected organ. Notably, mutant hepatocytes present an increase in mitochondrial ROS levels triggering mitochondrial nucleic acid release in the cytosol and type I interferon-mediate inflammation. In my opinion, the experiments are well designed and convincingly established the molecular mechanisms leading to the observed severe liver inflammation, which represent a central aspect of the pathological phenotype. However, although emphasis is placed on tissue specificity, it seems that the severity of the pathology correlates for most part with CIV levels/activity. Moreover, compared to the COX14 model, CIV is less affected in liver from the COA3 mutant mice, which present a milder phenotype. The reason why CIV accumulates at different levels in different organ/mice remains to be elucidated. Lastly, the authors observed the formation of extensive and numerous contact sites between mitochondria and lipid droplets in COX14 mutant hepatocytes. While it remains descriptive, this is an interesting observation that grants further investigation.

Minor points:

- It is very clear that lack of COX14 dramatically affects COX1 synthesis. However, the defect in newly synthesized COX1 stability is less convincing. In lane 4 of figure 1E, COX1 is undetectable, despite the lane appears overloaded, and it is difficult to assess its degradation. Please also indicate whether the mitochondria analyzed in figure 1D and E were isolated from liver. Additionally, the authors could comment on the stability of the other mitochondrion-encoded COX subunits, which stability has been reported to be compromised by defects in COX1 biogenesis.
- Treatment with NAC reduces mtRNA-dependent ISG up-regulation. Does the treatment ameliorate the liver pathology?
- Page 6, line 138. Supplemental fig. 1C should be supplemental fig. 1D.

Reviewer #2 (Remarks to the Author):

The authors discovered that the release of mitochondrial RNA is stimulated by elevated production of reactive oxygen species. This conclusion was reached through an analysis of two genetically modified mice, each carrying a mutation in either COX14 or COA3. Both of these mutations are related to the biogenesis of COX1, which is a component of complex IV. By studying these mice, the authors elucidated a new pathway in which the increased production of reactive oxygen species, resulting from the loss of complex IV, triggers the release of mitochondrial RNA. This process also leads to the simultaneous onset

of inflammation in the liver. These findings are highly intriguing as they indicate a strong connection between metabolic burdens like type 2 diabetes and steatohepatitis. The experiments were executed with great proficiency, and the interpretation is deemed satisfactory. In an effort to enhance the comprehensibility of this paper for the readers, there exist various remarks that necessitate further clarification.

- 1) Please provide an in-depth analysis of the results depicted in figures 1b, 1c, 5a, 5e, 5f, 7c, and 7d, highlighting the significance of these findings, as well as the individual data.
- 2) Please quantify all the results obtained from the Western blot analysis.
- 3) I believe it would be more advantageous to present more precise data in figure 2c rather than focusing on general changes of increase or decrease.
- 4) Fig. 5G lacks data on MAVS, despite the author's mention of it in the results section. It is worth noting the heightened presence of ZBP1 protein in the COX1 mutant.
- 5) An explanation is lacking for Figure 6.
- 6) Could you provide further insight regarding the liver phenotype's preeminence and whether COX14 expression exhibits variations across different tissues?
- 7) What is the underlying cause for the abbreviated lifespan observed in female mice with the COX14 mutation? Do disparities in liver phenotype exist between the sexes?

Reviewer #3 (Remarks to the Author):

This study addresses tissue-specific manifestations of mitochondrial diseases in two novel transgenic mouse models carrying patient-specific variants in COX14 and COA3, which are involved in the translation of cytochrome C oxidase (COX). Such studies are important because mitochondrial diseases are considered the largest group of inherited metabolic disorders, with more than >400 disease genes reported to date. Moreover, there are no effective therapies for these diseases and improved understanding of the molecular mechanisms may help in development of much needed treatments.

A long-standing question in the field has been why do mitochondrial diseases exhibit tissue-specific pathologies, when one might expect that the synthesis of ATP would be of importance to all tissues. Such tissue-specific manifestation does not seem to be explained simply by tissue-specific expression of disease-related genes as most are ubiquitously expressed and are considered typically essential for development and function in most organ systems.

During the last 5-10 years there has been a substantial amount of evidence published describing release of mitochondrial nucleic acids into the cytosol and activating the cellular antiviral signalling response

leading to inflammation including in monogenic mitochondrial disease (Dhir, A. et al. Mitochondrial double-stranded RNA triggers antiviral signalling in humans. *Nature* 560, 238–242 (2018). Here, Aich et al demonstrate that this pathway is activated in COX14 mouse tissues, in particular the liver, and to a lesser extent the COA3 liver. Treatment of COX14 hepatocytes with the antioxidant N-acetylcysteine for 24 hours attenuated accumulation of both mitochondrial RNAs in the cytosol and expression of inflammatory proteins. Therefore the main finding of this study and mechanistic insight concluded is that liver-specific inflammation is caused by a ROS-dependent release of mitochondrial nucleic acids into the cytosol.

In general experiments and data presented are to a high standard, the transgenic mice are well-phenotyped and manuscript is written well. The evidence supporting release of mitochondrial nucleic acids into the cytosol seems convincing but evidence for ROS involvement less so.

My specific comments:

- There is not a great deal of evidence to support the claim that the effects seen are ROS-dependent. Increased superoxide is shown in COX14 primary hepatocytes however I did not see anywhere the superoxide measurements following NAC treatment. Did NAC treatment cause a reduction in superoxide levels?
- Can NAC treatment work through other redox mechanisms than ROS that might also explain the effects seen?
- NAC only has a moderate effect on inflammatory mRNA gene expression (Fig 7C) but near complete reduction at protein level (Fig 7E).....and a complete reduction in mitochondrial mRNA abundance in cytosol (Fig 7D). An explanation to reconcile these observations is lacking.
- Apparently, another antioxidant, Mito-tempo, was used (ln 352) with similar effects to NAC but data is not shown.
- The liver-specific claim of inflammation seems to be a little overplayed, especially when inflammation is observed in other tissues at mRNA level and in kidney at protein level (and in my opinion also the spleen. Spleen has increased OAS1A. And in other tissues IFIT1 is not measured, so it doesn't seem to be entirely fair comparison).
- It isn't really clear from the discussion to what extent we learn about tissue-specific manifestation other than in the two models presented there is a bit more inflammation in the liver than in other tissues. This could be important but there isn't much in the discussion that puts this in the context of COX14 patients, and their multisystem disease, and mitochondrial diseases in general. In this regard, the final two concluding sentences of the discussion are somewhat at odds with the main message of the paper: "The concomitant loss of complex IV leads to increased ROS production, mitochondrial damage, and mtRNA release, which induces type I IFN inflammation in mice that affects not only liver but other major organ systems as well. Our findings provide a mechanistic explanation on how defective mitochondrial translation due to loss of COX14 function triggers ROS-induced type I IFN inflammation in liver leading to worsening pathology with time.

-

Minor comments

- It is not specified always which sex or age mice are used e.g. in biochemical analyses Figure 1 and Supp fig 1, Fig 2C.
- Fig 1C and Supp Fig 1B. COX14 protein amount is measured relative to WT. COX14 have been

normalised to VDAC for total protein for each sample but also ideally normalisation would be done to another mitochondrial protein to control for mitochondrial mass.

Fig 3a/b – some of the most significant changes are not annotated apparently because they do not belong to the relevant pathways highlighted. But it is not really clear until you come to Fig 3c which gives a summary of pathways. Perhaps order could be rearranged so that 3C comes first?

- Ln P48 – “absence of complex IV” would reduction or deficiency be more appropriate term?
- Ln 49 – “Additionally, we generated a COA3Y72C mouse, affected in COX1 biogenesis” – this is rather vague sentence, it would be good to state what COA3 is or does.
- Ln “M19I exchange” – substitution or missense variant
- Ln139 “Among the tested tissues, liver appeared to be the most affected tissue. In agreement with this observation, COX14M19I 140 mitochondria displayed a significant reduction in respiration (Fig. 1H).”
How can such agreement be postulated when only liver in OCR shown? Please rephrase.
- Ln 202 “Since the liver fulfills key metabolic functions, we addressed changes in the gene expression pattern by RNA sequencing (RNA-seq) of wild type and COX14M19I mice liver samples” - I don't see the logic in this statement, what about the other tissues, they are also important right? I get that not everything can be done. You chose the liver because it seems worse affected?
- Ln228 “Interestingly, many mitochondrial ribosomal proteins in this network map were upregulated” - Perhaps downregulated rather than downregulated. There is a mistake in the figure legend as both red and blue are said to be upregulated.

REVIEWER COMMENTS

Reviewer #1 (Remarks to the Author):

The assembly of cytochrome c oxidase (COX), the last enzyme of the mitochondrial respiratory chain, requires the function of numerous assembly factors, two of which, COX14 and COA3, are required for the expression and assembly of COX subunit 1. Mutations in both factors have been reported in patients with mitochondrial disorders, although with different severity. How mutations comparably affecting COX lead to different pathological manifestations remains to be fully elucidated. To gain insights into the pathophysiology of these disorders, in the manuscript by Aich et al, the authors generated and characterized two murine models recapitulating the COX14 and COA3 patient mutations. The mutation in COX14 is associated with multisystemic alterations in several tissue, with the liver being the most affected organ. Notably, mutant hepatocytes present an increase in mitochondrial ROS levels triggering mitochondrial nucleic acid release in the cytosol and type I interferon-mediate inflammation. In my opinion, the experiments are well designed and convincingly established the molecular mechanisms leading to the observed severe liver inflammation, which represent a central aspect of the pathological phenotype.

We thank the reviewer for the insights and comments to improve the manuscript. We have strived to address all concerns raised experimentally.

However, although emphasis is placed on tissue specificity, it seems that the severity of the pathology correlates for most part with CIV levels/activity. Moreover, compared to the COX14 model, CIV is less affected in liver from the COA3 mutant mice, which present a milder phenotype. The reason why CIV accumulates at different levels in different organ/mice remains to be elucidated.

Although a correlation with complex IV levels/activity exists, this appears to be not the only factor that contributes to the mitochondrial damage response. As we demonstrate below, e.g. ROS levels seem to be also tissue specifically altered. The topic of tissue specificity of mitochondrial dysfunction is an unresolved and very much investigated question in the field. Current concepts consider the mitochondrial turnover rate, what substrate are used by a given tissue to drive metabolism, and how easily each tissue regenerates as important aspects of the pathology. Accordingly, a plethora of factors add to the complexity of the tissue specificity and thus pathology. We have tried to make this clearer in the revised text.

As requested, we examined complex IV activity in different organs of the COA3 mice (new Fig 5d). Interestingly we observe a similar trend of reduced CIV activity in brain, muscle and liver. Yet, the COA3 mice did not display a significant CIV reduction in the heart.

The observed tissue phenotypes in liver and heart of COX14 mice correlate with complex IV activity. Brain and muscle display similar complex IV reduction,

yet a brain phenotype was not apparent in our analyses. Moreover, while we notice the most drastic reduction of the mutant COX14 protein in liver, one may argue that the reduced levels of COX14 in liver correlate with the complex IV defect. Yet, this correlation is not as apparent in the other tissues. Accordingly, as described above, there seem to be several factors that contribute to tissue specificity of mitochondrial disorders and the exact correlation between molecular defect and pathology remains an open and very much investigated issue in the field.

Lastly, the authors observed the formation of extensive and numerous contact sites between mitochondria and lipid droplets in COX14 mutant hepatocytes. While it remains descriptive, this is an interesting observation that grants further investigation.

To further investigate this, we isolated lipid body-associated and non-associated (cytosolic) mitochondria from WT and COX14^{M19I} mice livers (Supplementary Figure 5a). However, we found no protein candidates that differed significantly in these fractions and would explain the formation of the extensive contact sites (Supplementary Figure 5b). Additionally, we did not find functional difference in the respiratory capacities of the mitochondria (Supplementary Figure 5c). We included this data into the manuscript to provide a full report of the phenotypic analyses. Yet, the molecular detail of the altered cell biology remains elusive even though we extend the analysis significantly in the revised version.

Minor points:

- It is very clear that lack of COX14 dramatically affects COX1 synthesis. However, the defect in newly synthesized COX1 stability is less convincing. In lane 4 of figure 1E, COX1 is undetectable, despite the lane appears overloaded, and it is difficult to assess its degradation. Please also indicate whether the mitochondria analyzed in figure 1D and E were isolated from liver. Additionally, the authors could comment on the stability of the other mitochondrion-encoded COX subunits, which stability has been reported to be compromised by defects in COX1 biogenesis.

As requested, we have replaced Figure 1E with a higher exposed version from the same experiment. Here the COX1 is clearly visible and correlates with the quantification. In addition, a similar quantification was done for the COX2/COX3 band and it is represented bellow. There is no difference between the WT and Mutant.

- *Treatment with NAC reduces mtRNA-dependent ISG up-regulation. Does the treatment ameliorate the liver pathology?*

We would be happy to do this experiment. Yet, the experiment would require a specific permission. Due to political pressure to minimize animal experiments in the state of Lower Saxony and elsewhere in Germany, the permissions will take about nine months to get. Hence, we are unable to perform these analyses within a reasonable timeframe.

- *Page 6, line 138. Supplemental fig. 1C should be supplemental fig. 1D.*

We corrected the citation.

Reviewer #2 (Remarks to the Author):

The authors discovered that the release of mitochondrial RNA is stimulated by elevated production of reactive oxygen species. This conclusion was reached through an analysis of two genetically modified mice, each carrying a mutation in either COX14 or COA3. Both of these mutations are related to the biogenesis of COX1, which is a component of complex IV. By studying these mice, the authors elucidated a new pathway in which the increased production of reactive oxygen species, resulting from the loss of complex IV, triggers the release of mitochondrial RNA. This process also leads to the simultaneous onset of inflammation in the liver. These findings are highly intriguing as they indicate a strong connection between metabolic burdens like type 2 diabetes and steatohepatitis. The experiments were executed with great proficiency, and the interpretation is deemed satisfactory. In an effort to enhance the comprehensibility of this paper for the readers, there exist various remarks that necessitate further clarification.

1) Please provide an in-depth analysis of the results depicted in figures 1b, 1c, 5a, 5e, 5f, 7c, and 7d, highlighting the significance of these findings, as well as the individual data.

As requested, the data in Figure 1 has been updated with p-values for individual changes and indicated by a *. For the panels of qPCR data in Figures 5 and 7, due to limited figure spacing, we generated tables with the statistical analysis. They are added at the bottom of this document (See pages 9 - 18).

2) Please quantify all the results obtained from the Western blot analysis.

As requested, data obtained from Western blot analyses has been quantified and represented as bar graphs in the figures.

3) I believe it would be more advantageous to present more precise data in figure 2c rather than focusing on general changes of increase or decrease.

As requested, the actual change relative to WT is represented in revised Figure 2C and the raw data provided as Supplementary Table 1.

4) Fig. 5G lacks data on MAVS, despite the author's mention of it in the results section. It is worth noting the heightened presence of ZBP1 protein in the COX1 mutant.

We apologise for the error. The experiment tested for MDA5 and not MAVS; the mention of MAVS in the text was an error and has been corrected to be MDA5. As requested, we have mentioned the heightened presence of ZBP1 in the text.

5) An explanation is lacking for Figure 6.

As requested, we provide an explanation for Figure 6.

6) Could you provide further insight regarding the liver phenotype's preeminence and whether COX14 expression exhibits variations across different tissues?

We have extended the discussion on the tissue phenotypes as suggested by reviewer 1. As requested, we have determined COX14 protein levels (western blot) and gene expression (qPCR) across tissues (see below). Differences observed did not clearly correlate with the observed tissue phenotypes.

Figure a: Quantification of steady state protein levels of COX14 in wild type mouse heart, brain, liver, muscle, spleen, kidney and eye in relation to RIESKE.

qPCR:

Figure b: Quantification of mRNA levels of *cox14* in wild type mouse heart, brain, liver, muscle, spleen, kidney and eye in relation to the cytosolic mRNA *ddx6*

7) What is the underlying cause for the abbreviated lifespan observed in female mice with the COX14 mutation? Do disparities in liver phenotype exist between the sexes?

The liver phenotype is similar in both the sexes. What could be different is how inflammation is responded to systemically. Additionally, the physiological response is different amongst sexes, see changes in body weight (Fig. 2a). It could be that additional factors such as hormones, pregnancy, and ROS production during aging could lead to the effect on lifespan.

Reviewer #3 (Remarks to the Author):

This study addresses tissue-specific manifestations of mitochondrial diseases in two novel transgenic mouse models carrying patient-specific variants in COX14 and COA3, which are involved in the translation of cytochrome C oxidase (COX). Such studies are important because mitochondrial diseases are considered the largest group of inherited metabolic disorders, with more than >400 disease genes reported to date. Moreover, there are no effective therapies for these diseases and improved understanding of the molecular mechanisms may help in development of much needed treatments.

A long-standing question in the field has been why do mitochondrial diseases exhibit tissue-specific pathologies, when one might expect that the synthesis of ATP would be of importance to all tissues. Such tissue-specific manifestation does not seem to be explained simply by tissue-specific expression of disease-related genes as most are ubiquitously expressed and are considered typically essential for development and function in most organ systems.

During the last 5-10 years there has been a substantial amount of evidence published describing release of mitochondrial nucleic acids into the cytosol and activating the cellular antiviral signalling response leading to inflammation including in monogenic mitochondrial disease (Dhir, A. et al. Mitochondrial double-stranded RNA triggers antiviral signalling in humans. Nature 560, 238–242 (2018). Here, Aich et al demonstrate that this pathway is activated in COX14 mouse tissues, in particular the liver, and to a lesser extent the COA3 liver. Treatment of COX14 hepatocytes with the antioxidant N-acetylcysteine for 24 hours attenuated accumulation of both mitochondrial RNAs in the cytosol and expression of inflammatory proteins. Therefore the main finding of this study and mechanistic insight concluded is that liver-specific inflammation is caused by a ROS-dependent release of mitochondrial nucleic acids into the cytosol.

In general experiments and data presented are to a high standard, the transgenic mice are well-phenotyped and manuscript is written well. The evidence supporting release of mitochondrial nucleic acids into the cytosol seems convincing but evidence for ROS involvement less so.

My specific comments:

- There is not a great deal of evidence to support the claim that the effects seen are ROS-dependent. Increased superoxide is shown in COX14 primary hepatocytes however I did not see anywhere the superoxide measurements following NAC treatment. Did NAC treatment cause a reduction in superoxide levels?

As requested, we measured the superoxide levels after NAC treatments and add this experimental data as Fig. 7f. As expected, NAC treatment caused a reduction in superoxide levels.

- Can NAC treatment work through other redox mechanisms than ROS that might also explain the effects seen?

We clarify in the revised text that it is possible that NAC acts via other redox mechanisms through glutathion. Yet, since we obtain similar results with MitoTempo (Fig. 7g), we provide additional support indicating that ROS are the central mediator of mitochondrial damage and mtRNA release.

- NAC only has a moderate effect on inflammatory mRNA gene expression (Fig 7C) but near complete reduction at protein level (Fig 7E).....and a complete reduction in mitochondrial mRNA abundance in cytosol (Fig 7D). An explanation to reconcile these observations is lacking.

As requested, an explanation to reconcile these observations has been added into the main text.

- Apparently, another antioxidant, Mito-tempo, was used (ln 352) with similar effects to NAC but data is not shown.

As requested, the data is shown as Figure 7G.

- The liver-specific claim of inflammation seems to be a little overplayed, especially when inflammation is observed in other tissues at mRNA level and in kidney at protein level (and in my opinion also the spleen. Spleen has increased OAS1A. And in other tissues IFIT1 is not measured, so it doesn't seem to be entirely fair comparison).

As requested, the data for IFIT1 was added as Supplementary Figure 6. In addition we address the inflammatory phenotype across tissues in the revised discussion.

- It isn't really clear from the discussion to what extent we learn about tissue-specific manifestation other than in the two models presented there is a bit more inflammation in the liver than in other tissues. This could be important but there isn't much in the discussion that puts this in the context of COX14 patients, and their multisystem disease, and mitochondrial diseases in general. In this regard, the final two concluding sentences of the discussion are somewhat at odds with the main message of the paper: "The concomitant loss of complex IV leads to increased ROS production, mitochondrial damage, and mtRNA release, which induces type I IFN inflammation in mice that affects not only liver but other major organ systems as well. Our findings provide a mechanistic explanation on how defective mitochondrial translation due to loss of COX14 function triggers ROS-induced type I IFN inflammation in liver leading to worsening pathology with time.

As requested, we have tried to discuss the phenotype of the mice in the context of the human patient. Yet, keeping in mind that the patient died shortly after birth and the phenotype has not been fully assessed. Based on the reviewer's recommendation, we have modified these two sentences in the main text.

Minor comments

- It is not specified always which sex or age mice are used e.g. in biochemical analyses Figure 1 and Supp fig 1, Fig 2C.

We have added details to the figure legends.

- Fig 1C and Supp Fig 1B. COX14 protein amount is measured relative to WT. COX14 have been normalised to VDAC for total protein for each sample but also ideally normalisation would be done to another mitochondrial protein to control for mitochondrial mass.

COX14 has been normalised to RIESKE and not VDAC in the original analysis. We are grateful to the reviewer to address this aspect. We have corrected this in the revised legend. Additionally, as requested, we performed an additional normalisation to ATP5B and see the pattern to be the same (see graph below).

Fig 3a/b – some of the most significant changes are not annotated apparently because they do not belong the relevant pathways highlighted. But it is not really clear until you come to Fig 3c which gives a summary of pathways. Perhaps order could be rearranged so that 3C comes first?

We have made modifications as suggested.

- Ln P48 – “absence of complex IV” would reduction or deficiency be more appropriate term?

Changed accordingly.

- Ln 49 –“Additionally, we generated a COA3Y72C mouse, affected in COX1 biogenesis” – this is rather vague sentence, it would be good to state what COA3 is or does.

Changed accordingly.

- Ln “M19I exchange” – substitution or missense variant

Changed accordingly.

- Ln139 “Among the tested tissues, liver appeared to be the most affected tissue. In agreement with this observation, COX14M19I 140 mitochondria displayed a significant reduction in respiration (Fig. 1H).” How can such agreement be postulated when only liver in OCR shown? Please rephrase.

Changed accordingly.

- Ln 202 “Since the liver fulfills key metabolic functions, we addressed changes in the gene expression pattern by RNA sequencing (RNA-seq) of wild type and COX14M19I mice liver samples” I don’t see the logic in this statement, what about the other tissues, they are also important right? I get that not everything can be done. You chose the liver because it seems worse affected?

Changed accordingly.

- Ln228 “Interestingly, many mitochondrial ribosomal proteins in this network map were upregulated” - Perhaps downregulated rather than downregulated. There is a mistake in the figure legend as both red and blue are said to be upregulated.

Changed accordingly.

Figure 5A

Number of families	1
Number of comparisons per family	8
Alpha	0.05

Šidák's multiple comparisons test	Mean Diff.	95.00% CI of diff.	Below threshold?	Summary	Adjusted P Value
WT - COX14M19I					
Irf7	-6.973	-9.009 to -4.937	Yes	****	<0.0001
Isg15	-14.35	-16.39 to -12.32	Yes	****	<0.0001
Usp18	-2.633	-4.669 to -0.5974	Yes	**	0.0052
Ifi27	-3.307	-5.343 to -1.271	Yes	***	0.0003
Eif4e3	-1.457	-3.493 to 0.5793	No	ns	0.3072
Gbp3	-13.15	-15.19 to -11.12	Yes	****	<0.0001
Oasl1	-5.657	-7.693 to -3.621	Yes	****	<0.0001
Ifit1	-7.737	-9.773 to -5.701	Yes	****	<0.0001

Test details	Mean 1	Mean 2	Mean Diff.	SE of diff.	N1	N2	t	DF
WT - COX14M19I								
Irf7	1	7.973	-6.973	0.6976	3	3	9.996	32
Isg15	0.8833	15.24	-14.35	0.6976	3	3	20.57	32
Usp18	0.9733	3.607	-2.633	0.6976	3	3	3.775	32
Ifi27	0.97	4.277	-3.307	0.6976	3	3	4.74	32
Eif4e3	0.9667	2.423	-1.457	0.6976	3	3	2.088	32
Gbp3	1.09	14.24	-13.15	0.6976	3	3	18.85	32
Oasl1	0.95	6.607	-5.657	0.6976	3	3	8.108	32
Ifit1	1.01	8.747	-7.737	0.6976	3	3	11.09	32

Figure 5E

Number of families	1
Number of comparisons per family	5
Alpha	0.05

Šidák's multiple comparisons test	Mean Diff.	95.00% CI of diff.	Below threshold?	Summary	Adjusted P Value
WT - COA3Y72C					
Irf7	-5.455	-8.097 to -2.813	Yes	****	<0.0001
Isg15	-5.143	-7.784 to -2.501	Yes	****	<0.0001
Ifi27	-2.62	-5.262 to 0.02193	No	ns	0.0528
Gbp3	-4.49	-7.132 to -1.848	Yes	***	0.0003
Oasl1	-1.675	-4.317 to 0.9669	No	ns	0.3843

Test details	Mean 1	Mean 2	Mean Diff.	SE of diff.	N1	N2	t	DF
WT - COA3Y72C								
Irf7	1.048	6.503	-5.455	0.9637	4	4	5.661	30
Isg15	0.885	6.028	-5.143	0.9637	4	4	5.336	30
Ifi27	0.805	3.425	-2.62	0.9637	4	4	2.719	30
Gbp3	1	5.49	-4.49	0.9637	4	4	4.659	30
Oasl1	0.705	2.38	-1.675	0.9637	4	4	1.738	30

Figure 5F

Number of families	1
Number of comparisons per family	8
Alpha	0.05

Šidák's multiple comparisons test	Predicted (LS) mean diff.	95.00% CI of diff.	Below threshold?	Summary	Adjusted P Value
WT - COX14M19I					
18S	-0.1867	-2.956 to 2.582	No	ns	>0.9999
ACTB	0.2067	-2.562 to 2.976	No	ns	>0.9999
ND4	-3.933	-6.702 to -1.164	Yes	**	0.002
ND6	-2.02	-5.116 to 1.076	No	ns	0.4159
ATP6	-5.05	-7.819 to -2.281	Yes	****	<0.0001
ATP8	-3.32	-6.416 to -0.2241	Yes	*	0.0296
COX3	-3.51	-6.279 to -0.7410	Yes	**	0.0067
S12	-8.7	-11.80 to -5.604	Yes	****	<0.0001

Test details	Predicted (LS) mean 1	Predicted (LS) mean 2	Predicted (LS) mean diff.	SE of diff.	N1	N2	t	DF
WT - COX14M19I								
18S	1	1.187	-0.1867	0.942	3	3	0.1982	29
ACTB	1.117	0.91	0.2067	0.942	3	3	0.2194	29
ND4	1.04	4.973	-3.933	0.942	3	3	4.175	29
ND6	1.03	3.05	-2.02	1.053	3	2	1.918	29
ATP6	1.167	6.217	-5.05	0.942	3	3	5.361	29
ATP8	1.09	4.41	-3.32	1.053	3	2	3.152	29
COX3	1.117	4.627	-3.51	0.942	3	3	3.726	29
S12	1.01	9.71	-8.7	1.053	3	2	8.261	29

Figure 7C

Number of families	8
Number of comparisons per family	6
Alpha	0.05

Tukey's multiple comparisons test	Predicted (LS) mean diff.	95.00% CI of diff.	Below threshold?	Summary	Adjusted P Value
Calr					
WT vs. WT + NAC	-0.1767	-5.416 to 5.063	No	ns	0.9997
WT vs. COX14 ^{M19I}	-0.2533	-4.940 to 4.433	No	ns	0.9989
WT vs. cox14	-0.47	-5.156 to 4.216	No	ns	0.9934
WT + NAC vs. COX14 ^{M19I}	-0.07667	-5.316 to 5.163	No	ns	>0.9999
WT + NAC vs. cox14	-0.2933	-5.533 to 4.946	No	ns	0.9988
COX14 ^{M19I} vs. cox14	-0.2167	-4.903 to 4.470	No	ns	0.9993
Isg15					
WT vs. WT + NAC	0.04	-5.200 to 5.280	No	ns	>0.9999
WT vs. COX14 ^{M19I}	-15.54	-20.23 to -10.86	Yes	****	<0.0001
WT vs. cox14	-3.997	-8.683 to 0.6898	No	ns	0.1207
WT + NAC vs. COX14 ^{M19I}	-15.58	-20.82 to -10.34	Yes	****	<0.0001
WT + NAC vs. cox14	-4.037	-9.276 to 1.203	No	ns	0.1864
COX14 ^{M19I} vs. cox14	11.55	6.860 to 16.23	Yes	****	<0.0001
Gbp3					
WT vs. WT + NAC	-0.025	-5.265 to 5.215	No	ns	>0.9999
WT vs. COX14 ^{M19I}	-5.03	-9.716 to -0.3435	Yes	*	0.0308
WT vs. cox14	-2.253	-6.940 to 2.433	No	ns	0.5847
WT + NAC vs. COX14 ^{M19I}	-5.005	-10.24 to 0.2346	No	ns	0.0663
WT + NAC vs. cox14	-2.228	-7.468 to 3.011	No	ns	0.6761
COX14 ^{M19I} vs. cox14	2.777	-1.910 to 7.463	No	ns	0.4054
Ifi27					

WT vs. WT + NAC	0.01667	-4.670 to 4.703	No	ns	>0.9999
WT vs. COX14 ^{M19I}	-6.93	-11.62 to -2.244	Yes	**	0.0013
WT vs. cox14	-2.747	-7.433 to 1.940	No	ns	0.4151
WT + NAC vs. COX14 ^{M19I}	-6.947	-11.63 to -2.260	Yes	**	0.0013
WT + NAC vs. cox14	-2.763	-7.450 to 1.923	No	ns	0.4097
COX14 ^{M19I} vs. cox14	4.183	-0.5031 to 8.870	No	ns	0.0963
Ifit1					
WT vs. WT + NAC	0.02667	-4.660 to 4.713	No	ns	>0.9999
WT vs. COX14 ^{M19I}	-5.553	-10.24 to -0.8669	Yes	*	0.014
WT vs. cox14	-0.5333	-5.220 to 4.153	No	ns	0.9904
WT + NAC vs. COX14 ^{M19I}	-5.58	-10.27 to -0.8935	Yes	*	0.0134
WT + NAC vs. cox14	-0.56	-5.246 to 4.126	No	ns	0.989
COX14 ^{M19I} vs. cox14	5.02	0.3335 to 9.706	Yes	*	0.0312
Ifit3					
WT vs. WT + NAC	-0.1567	-4.843 to 4.530	No	ns	0.9997
WT vs. COX14 ^{M19I}	-3.327	-8.013 to 1.360	No	ns	0.2491
WT vs. cox14	-0.8767	-5.563 to 3.810	No	ns	0.96
WT + NAC vs. COX14 ^{M19I}	-3.17	-7.856 to 1.516	No	ns	0.2891
WT + NAC vs. cox14	-0.72	-5.406 to 3.966	No	ns	0.9771
COX14 ^{M19I} vs. cox14	2.45	-2.236 to 7.136	No	ns	0.5154
Oasl1					
WT vs. WT + NAC	0.46	-4.780 to 5.700	No	ns	0.9955
WT vs. COX14 ^{M19I}	-26.28	-30.97 to -21.60	Yes	****	<0.0001
WT vs. cox14	-11.81	-16.50 to -7.124	Yes	****	<0.0001
WT + NAC vs. COX14 ^{M19I}	-26.74	-31.98 to -21.50	Yes	****	<0.0001
WT + NAC vs. cox14	-12.27	-17.51 to -7.030	Yes	****	<0.0001
COX14 ^{M19I} vs. cox14	14.47	9.787 to 19.16	Yes	****	<0.0001
Usp18					
WT vs. WT + NAC	0.04667	-5.193 to 5.286	No	ns	>0.9999
WT vs. COX14 ^{M19I}	-7.423	-12.11 to -2.737	Yes	***	0.0005

WT vs. cox14	-1.973	-6.660 to 2.713	No	ns	0.6829
WT + NAC vs. COX14 ^{M19I}	-7.47	-12.71 to -2.230	Yes	**	0.0021
WT + NAC vs. cox14	-2.02	-7.260 to 3.220	No	ns	0.7389
COX14 ^{M19I} vs. cox14	5.45	0.7635 to 10.14	Yes	*	0.0164

Test details	Predicted (LS) mean 1	Predicted (LS) mean 2	Predicted (LS) mean diff.	SE of diff.	N1	N2	q	DF
Calr								
WT vs. WT + NAC	0.7933	0.97	-0.1767	1.982	3	2	0.1261	59
WT vs. COX14 ^{M19I}	0.7933	1.047	-0.2533	1.773	3	3	0.2021	59
WT vs. cox14	0.7933	1.263	-0.47	1.773	3	3	0.375	59
WT + NAC vs. COX14 ^{M19I}	0.97	1.047	-0.07667	1.982	2	3	0.05471	59
WT + NAC vs. cox14	0.97	1.263	-0.2933	1.982	2	3	0.2093	59
COX14 ^{M19I} vs. cox14	1.047	1.263	-0.2167	1.773	3	3	0.1729	59
lsg15								
WT vs. WT + NAC	1.07	1.03	0.04	1.982	3	2	0.02854	59
WT vs. COX14 ^{M19I}	1.07	16.61	-15.54	1.773	3	3	12.4	59
WT vs. cox14	1.07	5.067	-3.997	1.773	3	3	3.189	59
WT + NAC vs. COX14 ^{M19I}	1.03	16.61	-15.58	1.982	2	3	11.12	59
WT + NAC vs. cox14	1.03	5.067	-4.037	1.982	2	3	2.88	59
COX14 ^{M19I} vs. cox14	16.61	5.067	11.55	1.773	3	3	9.212	59
Gbp3								
WT vs. WT + NAC	0.66	0.685	-0.025	1.982	3	2	0.01784	59
WT vs. COX14 ^{M19I}	0.66	5.69	-5.03	1.773	3	3	4.013	59
WT vs. cox14	0.66	2.913	-2.253	1.773	3	3	1.798	59
WT + NAC vs. COX14 ^{M19I}	0.685	5.69	-5.005	1.982	2	3	3.571	59
WT + NAC vs. cox14	0.685	2.913	-2.228	1.982	2	3	1.59	59
COX14 ^{M19I} vs. cox14	5.69	2.913	2.777	1.773	3	3	2.215	59
lfi27								
WT vs. WT + NAC	0.7267	0.71	0.01667	1.773	3	3	0.0133	59
WT vs. COX14 ^{M19I}	0.7267	7.657	-6.93	1.773	3	3	5.529	59

WT vs. cox14	0.7267	3.473	-2.747	1.773	3	3	2.191	59
WT + NAC vs. COX14 ^{M19I}	0.71	7.657	-6.947	1.773	3	3	5.542	59
WT + NAC vs. cox14	0.71	3.473	-2.763	1.773	3	3	2.205	59
COX14 ^{M19I} vs. cox14	7.657	3.473	4.183	1.773	3	3	3.337	59
Ifit1								
WT vs. WT + NAC	0.6033	0.5767	0.02667	1.773	3	3	0.02127	59
WT vs. COX14 ^{M19I}	0.6033	6.157	-5.553	1.773	3	3	4.43	59
WT vs. cox14	0.6033	1.137	-0.5333	1.773	3	3	0.4255	59
WT + NAC vs. COX14 ^{M19I}	0.5767	6.157	-5.58	1.773	3	3	4.452	59
WT + NAC vs. cox14	0.5767	1.137	-0.56	1.773	3	3	0.4468	59
COX14 ^{M19I} vs. cox14	6.157	1.137	5.02	1.773	3	3	4.005	59
Ifit3								
WT vs. WT + NAC	0.06333	0.22	-0.1567	1.773	3	3	0.125	59
WT vs. COX14 ^{M19I}	0.06333	3.39	-3.327	1.773	3	3	2.654	59
WT vs. cox14	0.06333	0.94	-0.8767	1.773	3	3	0.6994	59
WT + NAC vs. COX14 ^{M19I}	0.22	3.39	-3.17	1.773	3	3	2.529	59
WT + NAC vs. cox14	0.22	0.94	-0.72	1.773	3	3	0.5744	59
COX14 ^{M19I} vs. cox14	3.39	0.94	2.45	1.773	3	3	1.955	59
Oasl1								
WT vs. WT + NAC	1.43	0.97	0.46	1.982	3	2	0.3282	59
WT vs. COX14 ^{M19I}	1.43	27.71	-26.28	1.773	3	3	20.97	59
WT vs. cox14	1.43	13.24	-11.81	1.773	3	3	9.422	59
WT + NAC vs. COX14 ^{M19I}	0.97	27.71	-26.74	1.982	2	3	19.08	59
WT + NAC vs. cox14	0.97	13.24	-12.27	1.982	2	3	8.756	59
COX14 ^{M19I} vs. cox14	27.71	13.24	14.47	1.773	3	3	11.55	59
Usp18								
WT vs. WT + NAC	0.5567	0.51	0.04667	1.982	3	2	0.0333	59
WT vs. COX14 ^{M19I}	0.5567	7.98	-7.423	1.773	3	3	5.922	59
WT vs. cox14	0.5567	2.53	-1.973	1.773	3	3	1.574	59
WT + NAC vs. COX14 ^{M19I}	0.51	7.98	-7.47	1.982	2	3	5.33	59

WT + NAC vs. cox14	0.51	2.53	-2.02	1.982	2	3	1.441	59
COX14 ^{M19I} vs. cox14	7.98	2.53	5.45	1.773	3	3	4.348	59

Figure 7D

Number of families	6
Number of comparisons per family	3
Alpha	0.05

Tukey's multiple comparisons test	Mean Diff.	95.00% CI of diff.	Below threshold?	Summary	Adjusted P Value
CALR					
WT vs. KO+NAC	-0.01333	-1.251 to 1.224	No	ns	0.9996
WT vs. KO	0.01	-1.228 to 1.248	No	ns	0.9998
KO+NAC vs. KO	0.02333	-1.214 to 1.261	No	ns	0.9988
COX2					
WT vs. KO+NAC	0.27	-0.9677 to 1.508	No	ns	0.8556
WT vs. KO	-3.423	-4.661 to -2.186	Yes	****	<0.0001
KO+NAC vs. KO	-3.693	-4.931 to -2.456	Yes	****	<0.0001
12S					
WT vs. KO+NAC	0.02667	-1.211 to 1.264	No	ns	0.9985
WT vs. KO	-3.07	-4.308 to -1.832	Yes	****	<0.0001
KO+NAC vs. KO	-3.097	-4.334 to -1.859	Yes	****	<0.0001
16S					
WT vs. KO+NAC	0.01	-1.228 to 1.248	No	ns	0.9998
WT vs. KO	-4.137	-5.374 to -2.899	Yes	****	<0.0001
KO+NAC vs. KO	-4.147	-5.384 to -2.909	Yes	****	<0.0001
ND1					
WT vs. KO+NAC	-0.003333	-1.241 to 1.234	No	ns	>0.9999
WT vs. KO	-4.47	-5.708 to -3.232	Yes	****	<0.0001
KO+NAC vs. KO	-4.467	-5.704 to -3.229	Yes	****	<0.0001
18S					
WT vs. KO+NAC	0.1533	-1.084 to 1.391	No	ns	0.9508
WT vs. KO	-0.23	-1.468 to 1.008	No	ns	0.8928
KO+NAC vs. KO	-0.3833	-1.621 to 0.8544	No	ns	0.7314

Test details	Mean 1	Mean 2	Mean Diff.	SE of diff.	N1	N2	q	DF
CALR								
WT vs. KO+NAC	1	1.013	-0.01333	0.5064	3	3	0.03724	36
WT vs. KO	1	0.99	0.01	0.5064	3	3	0.02793	36
KO+NAC vs. KO	1.013	0.99	0.02333	0.5064	3	3	0.06517	36
COX2								
WT vs. KO+NAC	1	0.73	0.27	0.5064	3	3	0.7541	36
WT vs. KO	1	4.423	-3.423	0.5064	3	3	9.561	36
KO+NAC vs. KO	0.73	4.423	-3.693	0.5064	3	3	10.32	36
12S								
WT vs. KO+NAC	1	0.9733	0.02667	0.5064	3	3	0.07448	36
WT vs. KO	1	4.07	-3.07	0.5064	3	3	8.574	36
KO+NAC vs. KO	0.9733	4.07	-3.097	0.5064	3	3	8.649	36
16S								
WT vs. KO+NAC	1	0.99	0.01	0.5064	3	3	0.02793	36
WT vs. KO	1	5.137	-4.137	0.5064	3	3	11.55	36
KO+NAC vs. KO	0.99	5.137	-4.147	0.5064	3	3	11.58	36
ND1								
WT vs. KO+NAC	1	1.003	-0.003333	0.5064	3	3	0.00931	36
WT vs. KO	1	5.47	-4.47	0.5064	3	3	12.48	36
KO+NAC vs. KO	1.003	5.47	-4.467	0.5064	3	3	12.47	36
18S								
WT vs. KO+NAC	1	0.8467	0.1533	0.5064	3	3	0.4282	36
WT vs. KO	1	1.23	-0.23	0.5064	3	3	0.6424	36
KO+NAC vs. KO	0.8467	1.23	-0.3833	0.5064	3	3	1.071	36

REVIEWER COMMENTS

Reviewer #1 (Remarks to the Author):

The authors have addressed the previous concerns raised by the reviewers and the revised manuscript is suitable for publication.

Reviewer #2 (Remarks to the Author):

The authors have adeptly addressed all of the reviewer's inquiries in a fitting manner.

While not a hindrance, for the Western blots in Fig 5b, 5g, 7e, and 7g, it is suggested to enhance reader comprehension by including molecular weight labels alongside the line-labeled markers—a considerate gesture for the reader.

Reviewer #3 (Remarks to the Author):

The authors have now included some data about MitoTEMPO, which is appreciated, but only offer some data about inflammatory gene expression. There is nothing about what MitoTEMPO did to the pathology (other than inflammation) and effects on ROS, which is central to the main claims of the study. ROS data was also missing from the NAC data in the original submission, which the authors now include, but could also easily have been included here for MitoTEMPO. However, I don't otherwise see any problem with the claim that ROS is mediating release of mtRNAs.

I still don't think there is much mechanistic insight gained about tissue-specificity, as is concluded in the abstract "Our study provides mechanistic insight into how defective mitochondrial gene expression causes tissue-specific inflammation. " Tissues with the most significant biochemical deficit (COX14 protein / cytochrome C oxidase) generally have more inflammation (As per Reviewer 1's comments). Just because some tissues don't correlate perfectly in this regard, it is probably more likely due to small sample size rather than a true finding. Also, COA3 mouse livers appear to induce inflammatory gene expression in liver even if other markers of liver pathology were subclinical in the COA3 model.

Figure 5 and Supp 6 is confusing as title refers to just one of the models but includes different types of data from the different models.

Reviewer #4 (Remarks to the Author):

General Comments

In the manuscript NCOMMS-23-19155A "Defective mitochondrial COX1 translation due to loss of COX14 function triggers ROS-induced inflammation in liver", the authors provide mechanistic insight on tissue

specific inflammation related to defective mitochondrial gene expression. The focus of my review is specifically regarding the lipidomics and proteomics data. I have serious concerns with the lipidomic data. The proteomic data is well done and I only have minor comments for the proteomics.

Specific Comments

1. The lipidomics method is highly unconventional and the method section is exceptionally weak. Typical quantitative lipidomics uses LC-MS and not just a direct infusion approach. I believe the QTRAP data was likely MRM acquired but that not fully indicated. Line 276 states the MS parameters are in S Table 1. However, S Table 1 appears to be the raw data? Upon further review, S Table 1 and S Table 2 are the same table. There are no MS parameters that I can find for either the QTRAP data or the Q Exactive data. Was the Q Exactive data LC-MS? How many biological replicates were analyzed? Since this experiment did not show lipid changes anyway, I suggest omitting it. If not omitted then this section requires significant details, it is impossible for me to evaluate the data if the methods are not clearly detailed.
2. The proteomic section is much more thorough and well written. A table with the TMT 6Plex labels vs sample would be helpful.
3. It would be more clear if the authors explicitly stated in a sentence that 3 biological replicates of WT and 3 biological replicates of the COX14 mutant were used. I infer that from the fact you used 6-Plex TMT, however, please clarify that in the methods.
4. Same comment for the mitochondrial proteomic samples. It is not declared how many samples are in the study and what is being compared.

REVIEWER COMMENTS

Reviewer #1 (Remarks to the Author):

The authors have addressed the previous concerns raised by the reviewers and the revised manuscript is suitable for publication.

Reviewer #2 (Remarks to the Author):

The authors have adeptly addressed all of the reviewer's inquiries in a fitting manner.

While not a hindrance, for the Western blots in Fig 5b, 5g, 7e, and 7g, it is suggested to enhance reader comprehension by including molecular weight labels alongside the line-labeled markers—a considerate gesture for the reader.

Has been done as requested.

Reviewer #3 (Remarks to the Author):

The authors have now included some data about MitoTEMPO, which is appreciated, but only offer some data about inflammatory gene expression. There is nothing about what MitoTEMPO did to the pathology (other than inflammation) and effects on ROS, which is central to the main claims of the study. ROS data was also missing from the NAC data in the original submission, which the authors now include, but could also easily have been included here for MitoTEMPO. However, I don't otherwise see any problem with the claim that ROS is mediating release of mtRNAs.

We have included the new data as requested (Figure 7g and New Supplemental Figure 8 g and h).

I still don't think there is much mechanistic insight gained about tissue-specificity, as is concluded in the abstract "Our study provides mechanistic insight into how defective mitochondrial gene expression causes tissue-specific inflammation. " Tissues with the most significant biochemical deficit (COX14 protein / cytochrome C oxidase) generally have more inflammation (As per Reviewer 1's comments). Just because some tissues don't correlate perfectly in this regard, it is probably more likely due to small sample size rather than a true finding. Also, COA3 mouse livers appear to induce inflammatory gene expression in liver even if other markers of liver pathology were subclinical in the COA3 model.

As indicated by the editor, we have tried to change the text accordingly and do not use the term "mechaistic" in the revised text. At severla places we state that the inflammatory severity correlates with the amount of complex IV reduction (e.g. "Accordingly, we reveal a pathology in which increased ROS production correlating with the loss of complex IV triggers mitochondrial RNA release and concomitant induction of inflammation pathways that contributes to hepatic failure.")

Figure 5 and Supp 6 is confusing as title refers to just one of the models but includes different types of data from the different models.

We have changed the titles.

Reviewer #4 (Remarks to the Author):

General Comments

In the manuscript NCOMMS-23-19155A “Defective mitochondrial COX1 translation due to loss of COX14 function triggers ROS-induced inflammation in liver”, the authors provide mechanistic insight on tissue specific inflammation related to defective mitochondrial gene expression. The focus of my review is specifically regarding the lipidomics and proteomics data. I have serious concerns with the lipidomic data. The proteomic data is well done and I only have minor comments for the proteomics.

Specific Comments

1. The lipidomics method is highly unconventional and the method section is exceptionally weak. Typical quantitative lipidomics uses LC-MS and not just a direct infusion approach. I believe the QTRAP data was likely MRM acquired but that not fully indicated. Line 276 states the MS parameters are in S Table 1. However, S Table 1 appears to be the raw data? Upon further review, S Table 1 and S Table 2 are the same table. There are no MS parameters that I can find for either the QTRAP data or the Q Exactive data. Was the Q Exactive data LC-MS? How many biological replicates were analyzed? Since this experiment did not show lipid changes anyway, I suggest omitting it. If not omitted then this section requires significant details, it is impossible for me to evaluate the data if the methods are not clearly detailed.

As requested, we clarified experimental details in the Materials and Methods section as well as expanded the Supplementary table 2. All data is now provided via MTBLS metabolights (see above). Lipidomics data is now available via Metabolights (MTBLS9823). Lipidomics reporting checklists are available via the following DOIs: [10.5281/zenodo.10891305](https://doi.org/10.5281/zenodo.10891305), [10.5281/zenodo.10891307](https://doi.org/10.5281/zenodo.10891307). Per standard, the Lipidomic Check list is part of the MetaboLight deposition and available online with the data (see DOIs above)

2. The proteomic section is much more thorough and well written. A table with the TMT 6Plex labels vs sample would be helpful.

We have included the information in the Material and Methods section; Labelling scheme for the comparison of cytoplasmic proteins in WT vs. COX14M19I has been included.

3. It would be more clear if the authors explicitly stated in a sentence that 3 biological replicates of WT and 3 biological replicates of the COX14 mutant were used. I infer that from the fact you used 6-Plex TMT, however, please clarify that in the methods.

We have clarified this as requested.

4. Same comment for the mitochondrial proteomic samples. It is not declared how many samples are in the study and what is being compared.

We modified the first sentence of the “Proteomics” section in the Materials and Method section accordingly. It now reads: “Hepatocytes from three biological replicates of WT and of COX14M19I were lysed ...”

REVIEWERS' COMMENTS

Reviewer #4 (Remarks to the Author):

The authors have addressed the previous concerns raised by the reviewers and the revised manuscript is suitable for publication.